# High-resolution micro-epidemiology of parasite spatial and temporal dynamics in a high malaria transmission setting in Kenya

Cody S. Nelson [1]*, Kelsey M. Sumner[2,3], Elizabeth Freedman[3], Joseph W. Saelens[3], Andrew A. Obala[4], Judith N. Mangeni[5], Steve M. Taylor[1,2,3,6] & Wendy P. O'Meara[1,3,6]

Novel interventions that leverage the heterogeneity of parasite transmission are needed to achieve malaria elimination. To better understand spatial and temporal dynamics of transmission, we applied amplicon next-generation sequencing of two polymorphic gene regions (*csp* and *ama1*) to a cohort identified via reactive case detection in a high-transmission setting in western Kenya. From April 2013 to July 2014, we enrolled 442 symptomatic children with malaria, 442 matched controls, and all household members of both groups. Here, we evaluate genetic similarity between infected individuals using three indices: sharing of parasite haplotypes on binary and proportional scales and the L1 norm. Symptomatic children more commonly share haplotypes with their own household members. Furthermore, we observe robust temporal structuring of parasite genetic similarity and identify the unique molecular signature of an outbreak. These findings of both micro- and macro-scale organization of parasite populations might be harnessed to inform next-generation malaria control measures.

[1] Duke Global Health Institute, Duke University, Durham, North Carolina, USA. [2] Department of Epidemiology, Gillings School of Global Public Health, University of North Carolina, Chapel Hill, North Carolina, USA. [3] Division of Infectious Diseases, Duke University School of Medicine, Durham, North Carolina, USA. [4] School of Medicine, Moi University College of Health Sciences, Eldoret, Kenya. [5] School of Nursing, Moi University College of Health Sciences, Eldoret, Kenya. [6] These authors contributed equally: Steve M. Taylor, Wendy P. O'Meara. *email: cody.nelson@duke.edu

The global burden of malaria has decreased considerably: from 2000 until 2015, cases declined by 41% and malaria deaths fell by 62%[1]. This improvement is broadly associated with the adoption of core control measures, principally the use of long-lasting insecticide-treated bednets, improved case management with rapid diagnostic tests (RDTs), and treatment with artemisinin-combination therapies (ACTs). Yet in some areas such as Bungoma county in western Kenya, *Plasmodium falciparum* transmission has failed to decline in proportion to control efforts, underscoring the need for new strategies[2,3]. Owing to this, there is renewed interest in developing, testing, and deploying supplemental strategies to accelerate malaria elimination[4], including enhanced surveillance, mass drug administration, and active community-based case detection[3]. Some of these strategies have been deployed in low/moderate transmission settings with variable success[5,6], but have been less frequently trialed in high-transmission settings of sub-Saharan Africa. Apart from the operational challenges of such settings, a major impediment to the application of these tools is a limited understanding of the fine-scale heterogeneity in malaria risk and transmission. Gaps in knowledge include: the extent to which time and geographic space structure parasite populations, the introduction or propagation of novel parasite strains, and the ability to identify hosts related through discrete parasite transmission chains. A greater understanding of the dynamics of natural infections and their impact on parasite transmissibility could enable rational implementation of control measures to reduce the malaria disease burden in high-transmission settings.

Next-generation sequencing (NGS) technologies may enable both local- and population-level tracking of parasite transmission[7]. However, *P. falciparum* molecular surveillance efforts have been hindered by: (1) the size and complexity of the parasite genome[8], (2) the high prevalence of polygenomic infections in high-transmission areas[9,10], and (3) the complex life cycle involving sexual recombination[11]. Collectively, these challenges limit the ability to define sequence identity of parasites in vivo, which is essential for identifying transmission between hosts. One potential solution is NGS of PCR amplicons, which enables resolution of parasite variants within a population into an array of genetically distinct haplotypes[12–14]. Using control parasite DNA mixtures, we[14–16] and others[13] have previously demonstrated that amplicon sequencing results in high-fidelity haplotype output with identity and frequency directly proportional to the input genetic material. The high sensitivity of this technique coupled with the ability to parse out multiple genotypes in polyclonal infection is highly appealing for the use of amplicon NGS data for malaria molecular epidemiology[7].

We previously conducted a case-control study of malaria in western Kenya in 2013–2014. Over a 15-month period, infected (case) and uninfected (control) index children along with all household members of both groups were enrolled and tested with malaria RDTs (study area map in Supplementary Fig. 1). We observed clustering of RDT-positive individuals, noting that infections were 2.5 times more common among the household members of cases compared to controls[2]. Though entomology in this study was quite limited, we also identified clustering of larval sites and bloodfed anopheline mosquitoes in case households; however, these relatively weak associations suggest that vector proximity is not a primary driver of disease risk in this context.

Here we employ parasites collected during this case-control study in western Kenya to interrogate malaria transmission across temporal and spatial scales. For this investigation, we utilize amplicon NGS of highly polymorphic *P. falciparum* genes, specifically circumsporozoite protein (*csp*) and apical membrane antigen-1 (*ama1*). Subsequently, we apply three metrics to assess interhost parasite genetic similarity: binary haplotype sharing (any haplotypes in common), proportional haplotype sharing (percentage of haplotypes in common), and the L1 norm (sequence-based distance). We hypothesized that this genotyping approach coupled with metrics of genetic similarity would yield information regarding spatial and temporal scales of malaria transmission that might be harnessed to target malaria control measures. Specifically, in this study we predicted that parasite populations in case children (CC) would be more genetically similar to asymptomatically infected members of their own households than to infected members of other households, and that the overall likelihood of genetically similar haplotypes between any two sampled individuals would be inversely related to geographic and temporal distance.

## Results

**Haplotype metrics and read coverage.** Of 5353 total study participants from across the study area (Fig. 1), 1050 were RDT+ for *P. falciparum*. A total of 966 RDT+ infections were submitted for amplicon NGS of *csp* and *ama1* loci (Fig. 1). After haplotype assignment[17] and quality filtering of reads, we identified 120 unique *csp* haplotypes across 655 participants and 180 *ama1* haplotypes across 667 participants (Fig. 1). In total, 617 samples (64% of infections initially submitted for sequencing) were assigned both *csp* and *ama1* haplotypes. Compared to un-genotyped samples, the median parasite density was nearly two orders of magnitude higher for samples successfully assigned *csp* (2.36 vs. 0.71, $p < 0.001$, Mann–Whitney U test) or *ama1* haplotypes (2.31 vs. 0.74, $p < 0.001$, Mann–Whitney U test) (Supplementary Table 1). In addition, a greater percentage of symptomatic children enrolled by passive case detection (CC) than asymptomatic household members were successfully assigned *csp/ama1* haplotypes (Supplementary Table 1), though this is likely a consequence of high parasite density in CC. Of participants successfully assigned *csp* haplotypes, 43.6% were CC, 38.4% case household members (CHMs), and 17.9% control household members (Table 1). Very similar data were obtained for samples assigned *ama1* haplotypes (Table 1). Furthermore, those assigned *csp* and *ama1* haplotypes were representative of the overall population of RDT+ study participants. The number of reads per participant was strongly correlated with $\log_{10}$ parasite density for both *csp* (Supplementary Fig. 2a; $\rho = 0.54$, $p < 0.001$, Spearman Rank test) and *ama1* (Supplementary Fig. 2b; $\rho = 0.47$, $p < 0.001$, Spearman Rank test). Across all successfully genotyped infections, the median read coverage was 13,369 for *csp* and 11,392 for *ama1* (Supplementary Fig. 2c, d).

**Multiplicity of infection.** Most study participants had polygenomic infections: single haplotypes were detected in only 34.7% (227/655; *csp*) and 33.9% (226/667; *ama1*) of genotyped infections (Supplementary Fig. 3a, b). The median number of haplotypes detected at each locus per study participant was 2 (Table 1, Supplementary Fig. 3a, b), with maxima of 16 (*csp*) and 14 (*ama1*). Overall intrahost nucleotide diversity was high, dominated by nonsynonymous sequence polymorphisms (Supplementary Fig. 4c, d). Of 227 participants with monogenomic infection at the *csp* locus, 58.1% also had only a single *ama1* haplotype (and vice versa—58.4%) (Supplementary Fig. 3c). Additionally, within individual participants, the number of *csp* and *ama1* haplotypes was highly correlated (Supplementary Fig. 3d; $\rho = 0.68$, $p < 0.001$, Spearman Rank test). However, there was no consistent difference between the MOI detected in CC vs. CHM within a single household (p = ns, Wilcoxon Sign-Rank test).

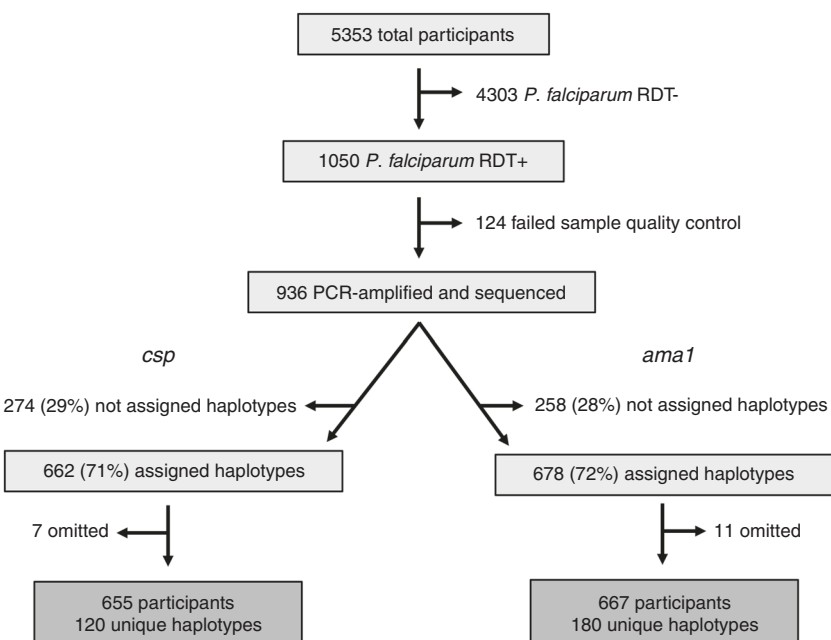

**Fig. 1 Study setup, sequencing, and haplotype calling.** All *P. falciparum* RDT+ subjects were selected for PCR amplification and sequence analysis. Amplicons within both *csp* and *ama1* hypervariable membrane proteins were sequenced. Approximately 30% of samples at both loci failed haplotype assignment due to low number of sequencing reads. *csp* sequencing data were excluded for seven study participants and *ama1* data for 11 participants due to either data inconsistencies or erroneous sample tracking/identification.

**Table 1 Descriptive statistics for study participants who were RDT+ and with successful *csp/ama1* haplotype assignment.**

| | | RDT+ ($n = 1050$) | Assigned *csp* haplotypes ($n = 662$) | Assigned *ama1* haplotypes ($n = 678$) |
|---|---|---|---|---|
| Person type | Case child | 43.1% | 43.6% | 43.2% |
| | Case household member | 39.0% | 38.4% | 39.3% |
| | Control household member | 17.0% | 17.9% | 17.4% |
| Median age (Range) | | 6 (0.08–82) | 6 (0.08–82) | 6 (0.08–82) |
| Median $\log_{10}$PD (Range) | | 2.16 (−0.86–6.67) | 2.36 (−0.51–6.67) | 2.31 (−0.86–6.67) |
| Median # haplotypes (Range) | | N/A | 2 (1–16) | 2 (1–14) |

**Case and haplotype distribution over time**. The distribution of participants with *csp* (Fig. 2a) and *ama1* (Supplementary Fig. 5a) haplotypes by month (April 2013 through June 2014) indicates year-round malaria transmission, though notable seasonal variation with case incidence peaking during the rainy season (approximately April through June). We observed heterogeneous haplotype persistence over time among the 120 *csp* (Fig. 2b) and 180 *ama1* haplotypes (Supplementary Fig. 5b): 4 *csp* and 2 *ama1* haplotypes were detected in at least 14 of 15 months, and a large proportion (48% of *csp* and 57% of *ama1*) appeared in only a single month (Table 2). The remaining haplotypes were detected intermittently over the study period, not necessarily during consecutive months.

We tested if haplotype presence was impacted by age, because parasite density (and thereby haplotype detection sensitivity) often depends upon host age in areas of endemic transmission[18,19]. To do so, we computed the prevalence difference (PD) of each haplotype ($PD_H$) between young children (≤5 years) and older children/adults (>5 years). However, we observed no consistent difference in haplotype prevalence between the ≤5 years and >5 years populations (Supplementary Fig. 6).

**Macro-level parasite genetic similarity**. We next investigated the overall temporal and spatial structuring of *csp* (Fig. 3) and *ama1*

(Supplementary Fig. 7) haplotypes. Visual inspection of the spatial distribution of haplotypes with variable duration (Table 2 —'persistent,' 'intermittent,' and 'sporadic') indicates that unique variants are dispersed across geographic space during the time period in which they are detected (Fig. 3a–c; Supplementary Fig. 7a–c). Furthermore, categorization of infections by administrative locations during the high-transmission season in 2013 (April–June) revealed that individual haplotypes were well mixed across the study area (Fig. 3d). Collectively, these findings suggest a lack of spatial structuring of haplotypes.

We subsequently employed binary sharing, proportional sharing, and the L1 norm, novel metrics which describe population-level genetic similarity, to examine *csp* and *ama1* genetic similarity over time (Fig. 3e–g; Supplementary Fig. 7d–f) and between administrative locations (Fig. 3h–m; Supplementary Fig. 7g–l). Binary sharing calculates whether a pair of infections share any haplotype, while proportional sharing expresses the pairwise comparison of haplotype sets as a proportion of the total number of haplotypes shared by infected individuals; the L1 norm is a unitless index of the diversity of sequences between any two populations, which were here defined as parasites constituting a single infection (see methods for details of calculation, Supplementary Fig. 8 for correlation of metrics). We computed each of these metrics for both *csp* and *ama1* haplotypes between all study months by iterative, random sampling of individuals from each

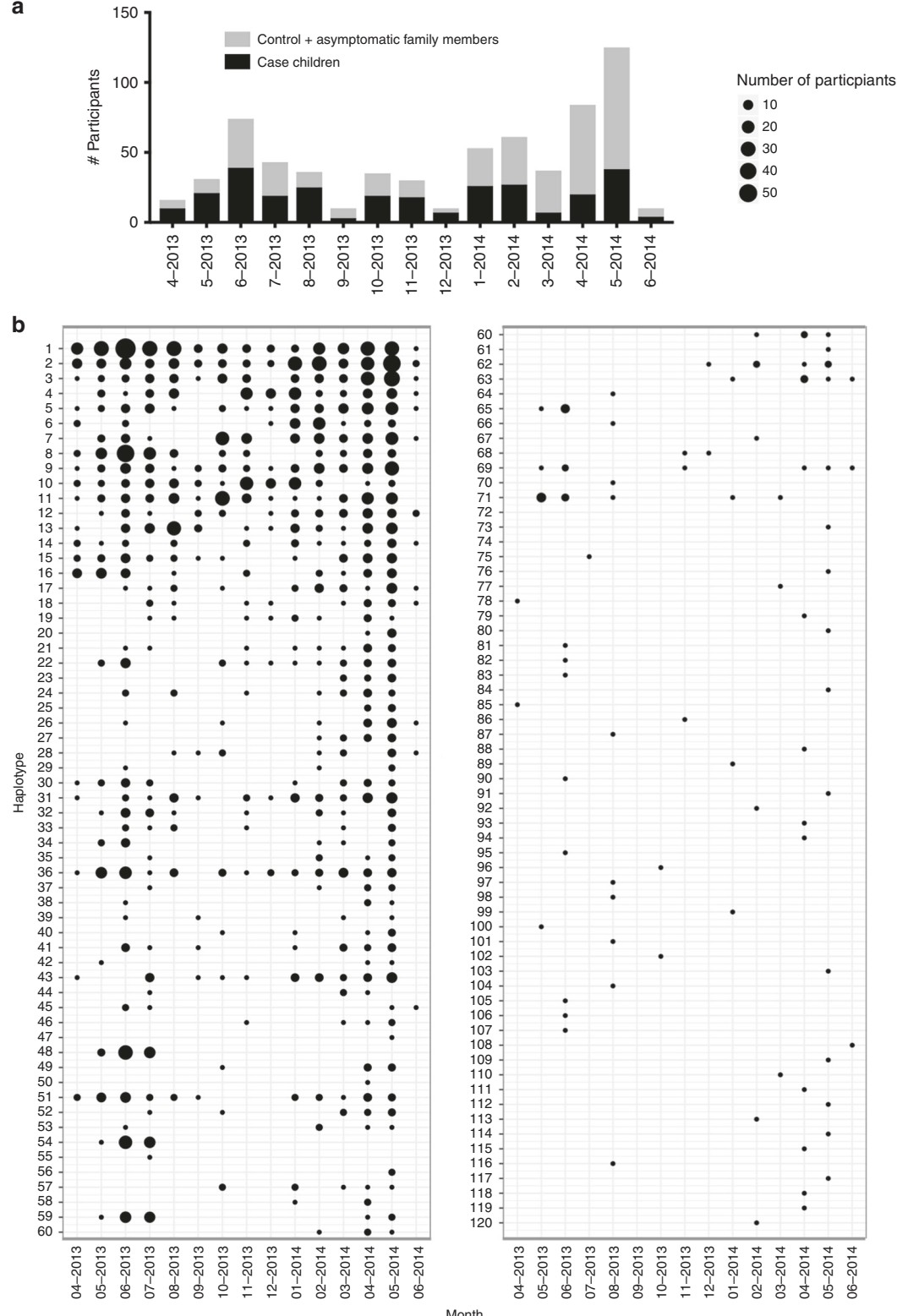

**Fig. 2 Heterogeneous persistence of *csp* haplotypes over time. a** Total number of study participants with *csp* haplotypes by month. Black denotes case children, and gray indicates both control and case household members. **b** Monthly prevalence of 120 unique *csp* haplotypes, sorted by overall prevalence. Size of circle indicates number of study participants with a particular haplotype in a given month.

month. Intriguingly, we identified a higher degree of parasite genetic similarity for individuals sampled within a single month compared with those sampled during different months (Fig. 3e–g; Supplementary Fig. 7d–f), which was statistically significant for binary sharing ($p = 0.029$, Mann–Whitney U test), proportional sharing ($p < 0.001$, Mann–Whitney U test), and the L1 norm ($p = 0.013$, Mann–Whitney U test) (Table S2). Analogous results were obtained for the temporal comparison of *ama1* haplotypes

**Table 2 Distribution of number of months unique *csp/ama1* haplotypes were detected.**

| Number of months present | Designation[a] | Number *csp* haplotypes | % *csp* haplotypes | Number *ama1* haplotypes | % *ama1* haplotypes |
|---|---|---|---|---|---|
| 15 | Persistent | 2 | 1.67 | 0 | 0 |
| 14 | | 4 | 3.33 | 2 | 1.11 |
| 13 | | 2 | 1.67 | 3 | 1.67 |
| 12 | Intermittent | 4 | 3.33 | 5 | 2.78 |
| 11 | | 5 | 4.17 | 8 | 4.44 |
| 10 | | 3 | 2.50 | 3 | 1.67 |
| 9 | | 1 | 0.83 | 6 | 3.33 |
| 8 | | 6 | 5.00 | 4 | 2.22 |
| 7 | | 3 | 2.50 | 4 | 2.22 |
| 6 | | 3 | 2.50 | 6 | 3.33 |
| 5 | | 5 | 4.17 | 3 | 1.67 |
| 4 | | 10 | 8.33 | 10 | 5.56 |
| 3 | Sporadic | 9 | 7.50 | 12 | 6.67 |
| 2 | | 5 | 4.17 | 12 | 6.67 |
| 1 | | 58 | 48.33 | 102 | 56.67 |

[a]Designation based upon number of study months each haplotype was detected: persistent >80% study months, intermittent 20% ≤ study months ≤ 80%, and sporadic < 20% study months

(Table S2). However, when genetic similarity was assessed for each combination of sampled administrative locations within a constrained window of time (highest malaria transmission season), we observed no consistent differences between metrics for pairs collected in the same location compared to pairs to those collected in different locations (Table S3). Overall, we observe spatially homogenous binary and proportional haplotype sharing (Fig. 3h–k; Supplementary Fig. 7g–j) though a heterogenous L1 norm (Fig. 3l, m; Supplementary Fig. 7k, l). Collectively, these findings indicate temporal structuring (though no clear spatial structuring) of parasite populations as defined by all metrics of genetic similarity at both *csp* and *ama1* loci.

**Micro- (household)-level parasite genetic similarity**. To investigate whether index cases and household members have similar parasite populations, and thus whether asymptomatic parasitemia among household members may be a risk factor for new clinical cases, we assessed genetic similarity between CC and asymptomatic CHMs from their household of origin. As a comparison, we also measured parasite genetic similarity between CC and members of an unrelated case household (URCHM), which was selected based on time of sampling (given the strong evidence for temporal macro-structuring of parasite populations). We evaluated CC:CHM and CC:URCHM genetic similarity using binary sharing, proportional sharing, and the L1 norm. Binary sharing was enhanced between CC and origin household CHM compared to binary sharing between CC and URCHM when computed using both *csp* (Fig. 4a; *p* = 0.02, Wilcoxon Signed-Rank test) and *ama1* haplotypes (Fig. 4b; *p* = 0.03, Wilcoxon Signed-Rank test). Similarly, proportional share scores were higher for CC with their origin CHM for both *csp* and *ama1* (Fig. 4c, d; *csp p* = 0.04, *ama1* *p* = 0.01, Wilcoxon Signed-Rank test). Finally, the L1 norm, which measures genetic distance between parasite populations, was statistically reduced for CC to origin CHM compared to URCHM at the *csp* (Fig. 4e; *p* = 0.03, Wilcoxon Signed-Rank test) but not *ama1* locus (Fig. 4f). These results were robust to alternative matching algorithms, including a combination of CC age, household location, and time of sampling. Lastly, if we lift the restriction that a household be comprised of 3+ individuals to be included in this analysis, the findings hold at the *csp* though not the *ama1* locus, possibly owing to a greater diversity of *ama1* haplotypes overall (180) compared to *csp* (120) and the resulting lower probability of observing exact matches.

We next examined the ability of genetic similarity metrics to predict the correct origin household for each CC from among houses with three or more infected household members (*n* = 38 households). To do so, we calculated aggregated *csp* and *ama1* binary sharing, proportional sharing, and the L1 norm between each pairwise combination of CC and CHM, yielding 1444 genetic relatedness values. We surmised that if binary sharing, proportional sharing, and the L1 norm are highly predictive indicies, the calculated CC:CHM value should be greatest for the comparison of CC with their own household members thus accurately identifying the CC household of origin. Overall, binary sharing was the most predictive of the origin household of a CC, with maximal binary share score correctly predicting the CC origin household 18% of the time (7/38) compared with 16% (6/38) for proportional sharing and 11% (4/38) for the L1 norm (Table 3). However, none of these were significantly different compared to what would be expected by random sampling alone (*p* = 0.06, 0.11, and 0.20, respectively, Fisher's Exact test). We also evaluated the ability of each sharing index to predict the time and geographic location of CC infections by comparing CC:CHM scores across time and space. All three genetic similarity metrics were highly predictive of CC temporal malaria acquisition, identifying the correct position in time of the CC ± 30 days (~15% maximal temporal distance) for approximately 50% of CCs (Table 3). All three indices were generally less predictive of CC spatial position, pinpointing the CC origin household within 2.25 km (~15% maximal geographic distance) for approximately 25% of CCs (Table 3).

**Outbreak: unique haplotype combination among CC**. To investigate the temporal and spatial relationship between parasite populations in CC, we computed pairwise binary sharing, proportional sharing, and L1 norm metrics for all CC infections (*csp* *n* = 283; *ama1* *n* = 288), then calculated the temporal (Fig. 5) and geographic (Supplementary Fig. 9) distance between CC pairings. When plotted against time between enrollment, we can clearly discern that *csp/ama1* binary and proportional sharing as well as parasite population sequence divergence (L1 norm) is highly dependent upon temporal distance between CC (Fig. 5). Furthermore, among geographically proximal infections (Supplementary Fig. 7), there is some enhanced binary sharing at both *csp* and *ama1* loci (Supplementary Fig. 9a, d) though no clear overall trend of increased genetic similarity.

Owing to the apparent temporal structuring of CC parasite haplotypes, we tested for specific clusters of parasite haplotypes in time among CC by computing the PD of each haplotype between CC and CHM (PD$_H$) (Fig. 6a, b) during each month of study enrollment. Comparing the monthly PD$_H$ of each haplotype, we determined that four haplotypes (*csp* H8, H48, and H54 as well as *ama1* H13) were significantly more common in CC than CHM during June 2013 (each *p* < 0.0001 by Fisher Exact test)

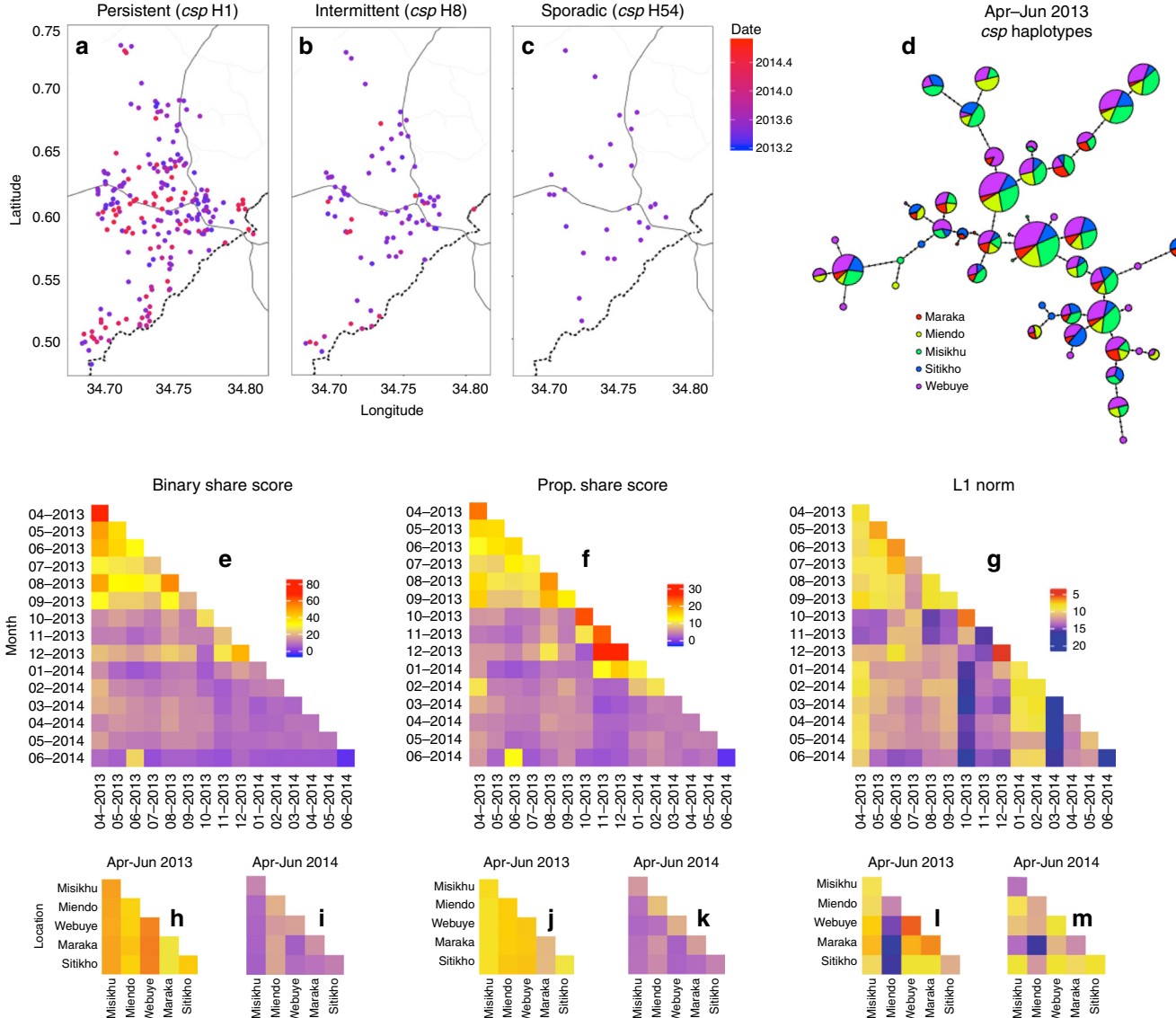

**Fig. 3 Genetic similarity of *csp* haplotypes is structured by time more than space. a–c** Location of study participants with 'persistent' haplotype *csp* H1 (**a**), 'intermittent' haplotype *csp* H8 (**b**), and 'sporadic' haplotype *csp* H54 (**c**). Blue color indicates the beginning (April 2013) and red the end (June 2014) of the study period, with the date denoting fractional years in decimal notation. **d** *csp* haplotype network for the high transmission season of April through June 2013 (for five most represented administrative locations: Maraka, Miendo, Misikhu, Sitikho, and Webuye). Each circle indicates a unique haplotype with size proportional to the log$_2$-scaled haplotype prevalence and color denoting the fractional prevalence in each administrative location. Haplotype connections were calculated using an infinite site model (Hamming distance) of DNA sequences, with dots along connections indicating the number of base-pair differences between sequences. **e–g** Temporal comparison heat maps of mean binary haplotype sharing (**e**), proportional haplotype sharing (**f**), and L1 norm genetic distance (**g**) calculated between months of study enrollment. **h–m** Spatial comparison heat maps of binary haplotype sharing (**h**, **i**), proportional haplotype sharing (**j**, **k**), and L1 norm genetic distance (**l**, **m**) calculated for a distinct temporal window (**h**, **j**, **l**: April-June 2013; **i**, **k**, **m**: April-June 2014) for the five most represented administrative locations (see above), which are arranged from north to south (see map in Fig. S1). For (**e–m**), blue denotes the minimum for each genetic similarity index, red the maximum, and yellow the midpoint.

(Fig. 6a, b). We examined CC from 5/6/2013–7/29/2013, noting all those infected with parasites bearing *csp* H8, H48, and H54 also had evidence of *csp* H1 (Fig. 6c). Likewise, from 5/6/2013–7/29/2013 all CC in which *ama1* H13 was detected also had *ama1* H5 and H8 (Fig. 6d). In total we identified 26 CC with *csp* H1 + H8 + H48 + H54 and 27 with *ama1* H5 + H8 + H13 (Fig. 6c, d). Intriguingly, we observed substantial overlap of this set of haplotypes in CC: 23 CC had evidence of all haplotypes combined (*csp* H1 + H8 + H48 + H54 and *ama1* H5 + H8 + H13) (Fig. 6e). In comparison, from 5/6/2013–7/29/2013 no CHM had either of these unique haplotype combinations (Fig. 6f, g). Thus, by comparing haplotype prevalences in CC and CHM, we identified

a combination of haplotypes that co-occur within a temporally restricted window and are associated with clinical disease.

Interestingly, we observe that this unique combination of *csp* and *ama1* haplotypes largely occurred in CC during a 3-week period from 6/17/2013 to 7/8/2013, peaking during the week of June 24th (Fig. 6h). We defined an outbreak 'case' as the presence of five or more of the seven outbreak haplotypes (*csp* H1/H8/H48/H54 and *ama1* H5/H8/H13), comprising more than 98% of the reads detected in an individual. Employing this definition, we identified a total of 29 outbreak cases and 48 non-outbreak cases among the 77 total CC between 5/6/2013 and 7/29/2013. We see a clear peak of genetically homogenous outbreak cases that

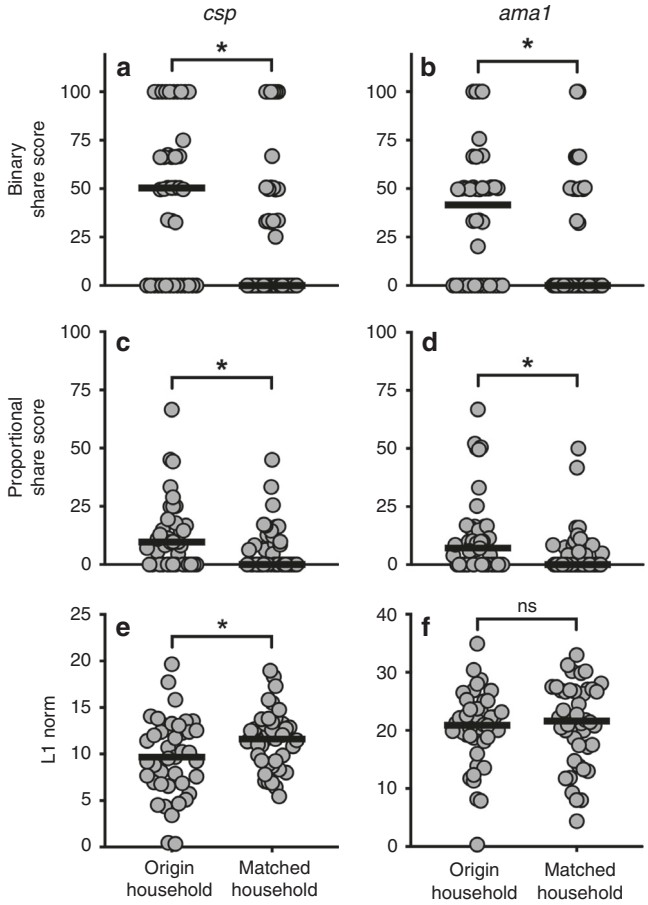

**Fig. 4 Enhanced parasite genetic similarity between case children and their own household members.** Genetic similarity metrics (binary share score, proportional share score, and L1 norm) were computed for case children (CC) with both their household of origin as well as with an unrelated household matched on time. Each metric was computed independently for *csp* and for *ama1*. **a, b** CC had significantly higher *csp* (**a**) and *ama1* (**b**) binary share scores with infected members of their origin households compared to members of matched households. **c, d** Furthermore, a higher proportional share score was observed for CC with their household of origin at both *csp* (**c**) and *ama1* (**d**) loci. **e, f** A reduced L1 norm for origin household (i.e. greater sequence similarity) was observed for *csp* (**e**) though not *ama1* (**f**) haplotypes. *$p < 0.05$, Wilcoxon Signed-Rank test.

accounts for nearly all cases during this 3-week period and is book-ended with endemic transmission of non-outbreak cases (Fig. 6i). Intriguingly, this outbreak event was not geographically confined, but rather was widely disseminated across the study area (Fig. 6j). The seemingly random geographic distribution was supported by a test for spatial structure of the outbreak haplotype combination in SaTScan, which identified 0 high clusters and 1 low cluster (Relative Risk: 0.00, $p = 0.71$, Bernoulli spatial scan statistic)—a null result.

## Discussion

In this study, we utilized amplicon deep sequencing of two *P. falciparum* polymorphic gene targets in parasites collected from households in western Kenya to investigate the geographic and temporal structuring of parasite populations. Our analyses indicate that temporally proximate infections have enhanced genetic similarity, suggesting that parasite populations are at least partially structured by time. Furthermore, we report that children

with malaria are more likely to share parasite haplotypes with asymptomatically infected members of their own household compared with members of time-matched households. Finally, we utilized temporal structuring of parasite populations to identify the genetic signature of an outbreak of parasite genotypes manifest as enrichment of specific parasite haplotype combinations during a single month in 2013. Collectively, our analysis identifies both micro- and macro-level organization of parasite populations, enhancing our understanding of malaria transmission heterogeneity.

To the best of our knowledge, this investigation presents the first genetic data directly linking parasites causing asymptomatic infections among household members with those causing symptomatic disease. Similar to prior studies[20–24], we previously noted that CC in this cohort were 2.5 times more likely to have RDT+ household members than were control children[2] indicating household-level hotspots of high-risk individuals. Herein, we extended this association using our parasite genotyping approach to quantify the degree of genetic similarity between infected individuals. For example, symptomatic children had a much higher median probability of sharing a *csp* haplotype with asymptomatic members of their own household (50%) than with those of a matched household (0%; $p = 0.02$, Wilcoxon Sign-Rank test), indicating that parasites infecting symptomatic children are more genetically similar to those in their own household than to the population at large (Fig. 4). These findings expand upon the prior recognition of household-clustering of malaria risk by providing direct evidence that symptomatic children are participating in the same parasite transmission network as their household members. What remains unknown is the direction of this transmission, because we cannot assess whether specific haplotype-identical parasites were transmitted from infected household members to CC, or whether both CC and household members acquired haplotype-identical parasites from the same source. Better understanding this phenomenon, through further high-resolution investigations into transmission networks within individual households, will be critical to discern whether treatment of asymptomatic infections might reduce the burden of clinical malaria disease.

Our amplicon NGS approach detected measurable temporal structuring of parasite populations in this geographically restricted, high-transmission setting, though no spatial structuring beyond the household unit. First, we observed temporal clusters of increased genetic similarity, with enhanced average similarity between the infections in any given month (Supplementary Table 2, Fig. 3, Supplementary Fig. 7) but not in any given location (Supplementary Table 3). Furthermore, genetic similarity metrics assessed between CC and household members was reliably more predictive of household temporal proximity (Table 3). Lastly, among CC we observed a strong inverse relationship between temporal distance and binary/proportional sharing (direct relationship between temporal distance and the L1 norm) (Fig. 5). Thus, all investigations point to a strong time-dependency of parasite transmission in this endemic setting, particularly among CC. Intriguingly, individual haplotypes (Fig. 3, Supplementary Fig. 7) and even unique haplotype combinations (Fig. 6) appear evenly distributed across the study area at any single point in time. The mechanism of this long-range transmission (cases up to 20 km apart) of genetically identical infections is perplexing, since the flight range of unfed *Anopheles gambiae* has been measured at a maximum of 3 km[25]. Nonetheless, this robust temporal structuring of haplotypes has profound implications for malaria elimination in high-transmission settings, suggesting that strategies focused purely on cluster-based control/prevention may fail to prevent forward parasite transmission and disease.

| | | BSS %[a] | BSS p-value[b] | PSS %[a] | PSS p-value[b] | L1 %[a] | L1 p-value[b] |
|---|---|---|---|---|---|---|---|
| Household of origin | | 18.4 | 0.06 | 15.7 | 0.11 | 10.5 | 0.20 |
| Temporal distance from origin household (% maximal temporal distance) | ±10 days (5%) | 42.1 | **<0.01** | 42.1 | **<0.01** | 34.2 | **<0.01** |
| | ±30 days (15%) | 50.0 | **0.03** | 50.0 | **0.03** | 55.3 | **<0.01** |
| | ±60 days (30%) | 60.5 | **0.02** | 60.5 | **0.02** | 65.8 | **<0.01** |
| Geographic distance from origin household (% maximal geographic distance) | within 0.75 km (5%) | 18.4 | 0.15 | 21.1 | 0.09 | 13.2 | 0.43 |
| | within 2.25 km (15%) | 26.3 | 0.40 | 26.3 | 0.40 | 23.7 | 0.57 |
| | within 4.5 km (30%) | 36.8 | 0.81 | 36.8 | 0.81 | 39.4 | 0.63 |

**Table 3 Maximal genetic similarity metrics are highly predictive of case child temporal distance from origin household.**

[a]Percentage of case children (n = 38) for which maximal binary share score, proportional share score, and L1 norm correctly predict case child household of origin OR temporal distance from origin household (within three temporal windows: ±10, 30, or 60 days representing 5, 15, or 30% of the maximum time between cases) OR geographic distance from origin household (within three ranges – 5, 15, and 30% maximal geographic distance)
[b]Fisher's Exact Test p-value based on assumption that cases are evenly distributed throughout time and across geographic space

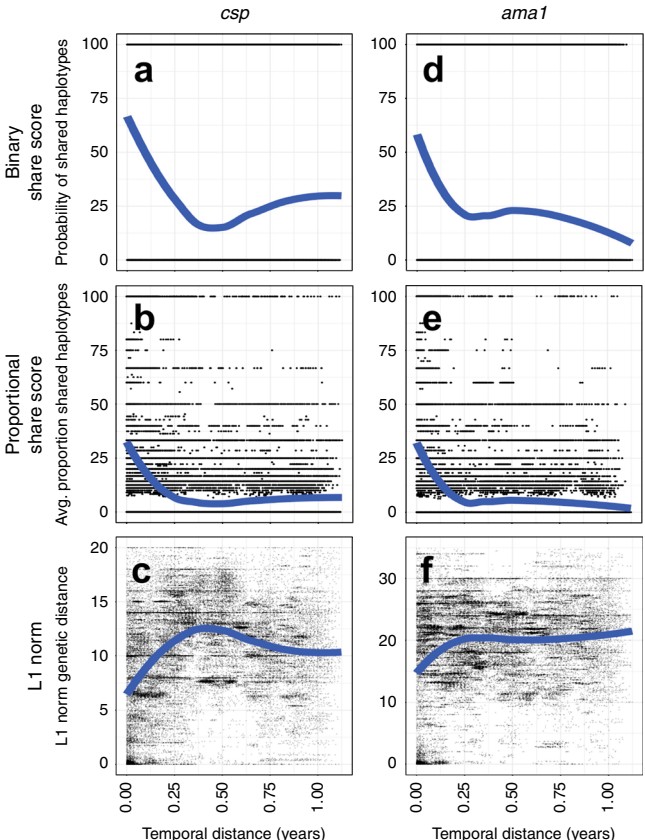

**Fig. 5 Case children with temporally proximal infections have enhanced genetic similarity of csp/ama1 haplotypes.** Genetic similarity metrics were computed for all possible pairings of case children for csp (n = 273) and ama1 (n = 288) haplotypes. **a**–**c** csp haplotype binary sharing (**a**), proportional sharing (**b**) and L1 norm (**c**) metrics for all CC pairwise comparisons is plotted against temporal distance for CC. **d**–**f** ama1 haplotype binary sharing (**d**), proportional sharing (**e**), and L1 norm (**f**) metrics for all CC pairings is plotted against temporal distance for CC. Blue lines indicate the locally estimated scatterplot smoothing (LOESS) regression fit of data.

Notably, this is the first study to detect the genetic signature of an outbreak amidst endemic malaria transmission. While several investigations have recognized genetic homogeneity in small-scale outbreaks[26,27] and traced the spread of drug-resistance alleles[28], we identified strong genetic evidence of a malaria outbreak nested within the temporal structure of haplotype sharing among CC, namely the appearance in symptomatic children of parasites with identical csp and ama1 combinations of haplotypes nearly simultaneously across the study area. As has been suggested[7] these ensembles of parasites, when detected in linked individuals, provide compelling evidence of a shared origin of infections. Some outbreak-associated haplotypes (csp H1/H8 and ama1 H5/H8) were exceedingly common throughout the study period and present in both CC and CHM during the majority of months, while other outbreak haplotypes (csp H48/H52 and ama1 H13) are rare and appear almost exclusively among CC from May–July 2013. One CC infected with all seven csp/ama1 outbreak haplotypes reported a travel history to another malaria-endemic region within the past 3 months, so we might hypothesize that rare haplotypes (csp H48/H54 and ama1 H13) were imported from outside the study area. The reason for the co-occurrence of this unique combination of haplotypes among symptomatic children is unclear, and two major questions remain unanswered by this investigation: (1) How did the outbreak spread nearly simultaneously across a relatively large geographic area and (2) why was this combination only detected among CC? The geographic co-occurrence of outbreak cases may be contingent upon unconventional and/or undescribed vector movement and biting behavior or cryptic human movements within the study area. Strain-specific immune response have been associated with protection from disease[29]—including those directed against a vaccine-elicited csp haplotype[13,30]—and therefore, it is possible that rare haplotypes represent new, antigenically diverse malaria strains to the study area (not previously encountered by CC) leading to enhanced clinical disease. Alternatively, these rare haplotypes could be genomically linked with novel virulence factors. Though we cannot discern the origin of vector transmission and/or cause of pathogenesis with any degree of certainty using our amplicon NGS data, the ability to detect and monitor genetically identical polyclonal infections over time demonstrates the power of this amplicon NGS sequencing approach for malaria molecular epidemiology.

Our sampling frame and genetic analytic approaches enabled us to detect signatures of parasite population structure in a high-transmission setting. While malaria cases are recognized to be clustered spatiotemporally[31–37], prior molecular epidemiology studies have generally failed to observe parasite population genetic structure (principally by geographic location) using a variety of genotyping methods, sampling schemes, and analytic tools[38–43]; one recent notable exception found a significant decay in genetic relatedness as a function of increasing distance using microsatellite genotyping of polygenomic infections, but reported from a low-transmission setting in Namibia[44]. In our high-transmission setting, we observed that parasite populations were structured (1) by household and (2) by time more than geography by the application of three novel metrics (binary sharing,

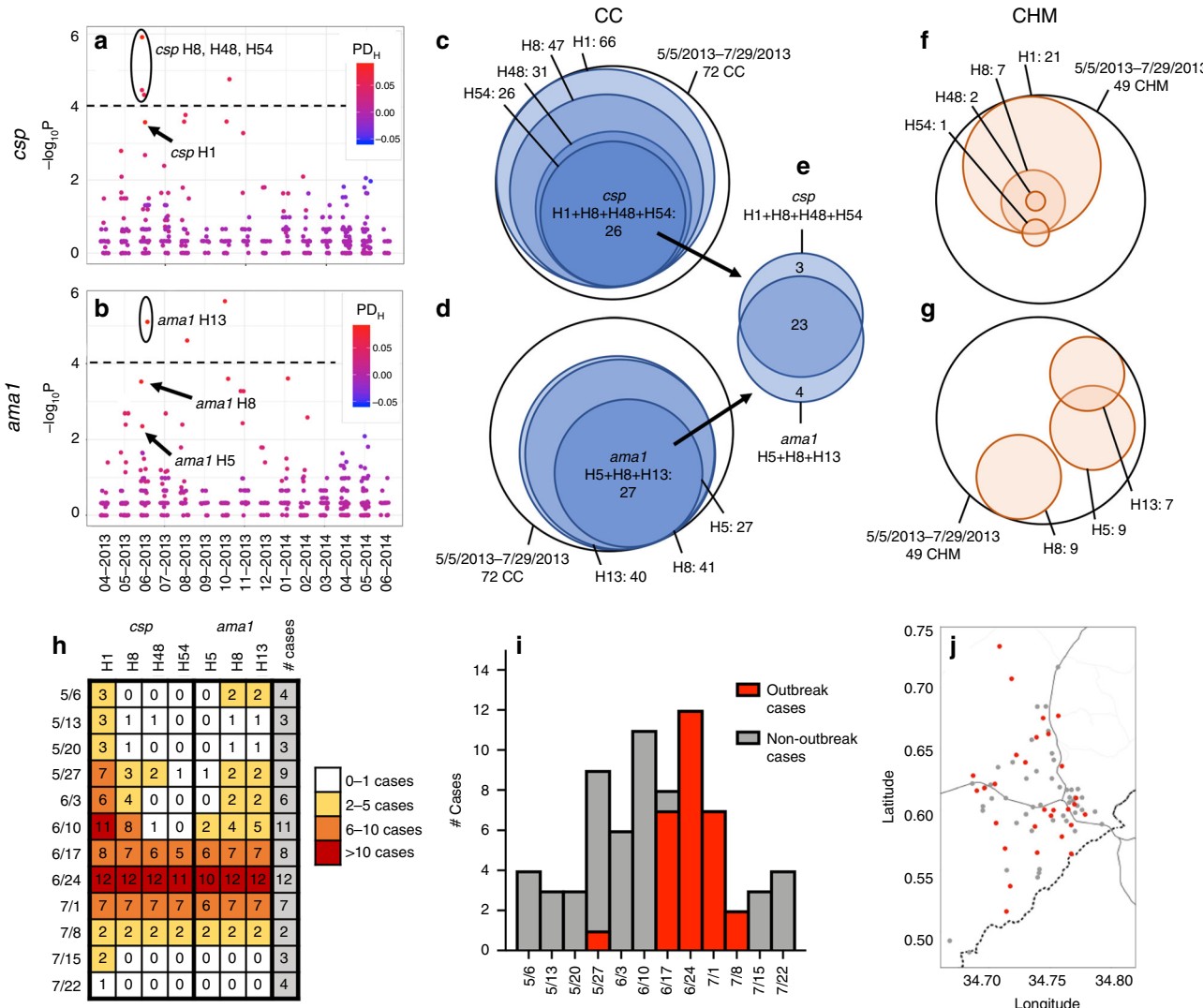

**Fig. 6 Discovery of unique haplotype combination in case children indicating malaria outbreak. a, b** The haplotype prevalence difference ($PD_H$) between case children (CC) and case household members (CHM) during each month was calculated for *csp* (**a**) and *ama1* (**b**). Color indicates $PD_H$, with red identifying haplotypes more common in CC and blue more common in CHM. The y-axis indicates the Fisher's Exact test $-\log_{10}$(p-value) for haplotype prevalence in CC vs. CHM, while the dotted line denotes the Bonferroni-corrected threshold for statistical significance. **c–g** Venn diagrams of *csp* and *ama1* haplotypes with a high incidence in case children for 72 CC and 49 CHM between 5/6/2013 and 7/28/2013 (only CC with both *csp* and *ama1* haplotypes included). **c, f** 26 CC had evidence of co-infection with *csp* haplotypes 1, 8, 48, and 54, compared to no CHM. **d, g** 27 CC had evidence of co-infection with *ama1* haplotypes 5, 8, and 13, compared to no CHM. **e** 23 CC were infected with CSP haplotypes 1, 8, 48, and 54 in addition to *ama1* 5, 8, and 13. **h** Number of CC each week from 5/6/2013–7/29/2013 infected with each haplotype associated with outbreak infection. White indicates 0–1 CC, yellow 2–5 CC, orange 6–10 CC, and red >10 CC infected with a given haplotype in a particular week. **i** Outbreak epidemiology curve. Weekly case incidence from 5/6/2013–7/28/2013 is shown by overall bar height. Weekly incidence of outbreak cases (defined as presence of >=5/7 outbreak haplotypes CSP H1/H8/H48/H54 and AMA H5/H8/H13, with these haplotypes comprising >98% reads detected) is indicated in red, and non-outbreak cases in gray. **j** Household location of case children (5/6/2013–7/28/2013) with outbreak cases (red) and non-outbreak cases (gray).

proportional sharing, and the L1 norm) to systematically interrogate the genetic similarity of polygenomic parasite populations present within individual hosts expressed as haplotypes of unlinked parasite genes *csp* and *ama1*. The concordant results from parallel analyses using these unlinked gene targets extends the credibility and utility of our novel metrics as tools. Two of these metrics, binary sharing (any shared haplotypes) and proportional sharing (percentage of shared haplotypes), express the genetic relationship between two infections as a function of the presence or absence of identical individual haplotypes within the pair. The premise of these metrics is that haplotypes that are identical by state represent parasites identical by descent; although this assumption would not be reasonable over large

distances or time periods owing to a variety of biological and epidemiologic factors, the constrained temporal and spatial scales and dense sampling of our study render this a more plausible assumption. The third metric, the L1 norm, has been used in viral genomic epidemiology as a measure of the sequence-based genetic divergence of two pathogen populations[45]. The polygenomic nature of most infections in high-transmission settings has precluded the use of numerous traditional tools of population genetics, but the use of our metrics enabled us to exploit these polygenomic infections, by capturing and then linking diverse haplotypes in separate hosts. We suggest that binary and proportional sharing metrics produce highly similar results in our analyses, whereas the L1 norm results are somewhat distinct and

heterogeneous (Fig. 3, Supplementary Figs. 7, 8). We anticipate that appropriate use of these metrics will depend on local epidemiology, namely parasite genetic diversity, transmission intensity, and prevalence of parasitemias. Thus, we propose these metrics ought to be applied to diverse datasets to define the context of their utility. Nevertheless, we hypothesize these high-resolution genetic metrics will enable investigators to identify connectivity between polygenomic infections on more granular temporal and geographic scales.

What are the implications of our findings for transmission-reducing strategies? The identification of similar parasite populations between CC and their household of origin supports notions in the malaria transmission field that symptomatic malaria infection may be partially fueled by asymptomatic infection of household members. Yet we also present strong evidence for temporal structuring of parasite populations and episodic transmission of genetically identical parasite haplotypes, unbounded by household or geography, suggesting that intra-household sharing is not the only source of new infections. At face value these findings are contradictory, which emphasizes the complexity of local and population-level transmission networks. Nonetheless, our findings are most relevant to reactive programs, in which index cases trigger either ring-testing (as in RACD) or ring-treatment (as in ring focal drug administration) in an effort to mitigate foci of transmission. These programs are popular and literature supports their efficacy at identifying additional cases, though there are insufficient data regarding cost-effectiveness and the impact of transmission intensity or diagnostic test sensitivity for these methods to be fully embraced. This investigation provides direct evidence that clinically silent parasite transmission chains within households are an important risk factor (but not the exclusive source) of new infections, which supports the rationale for employing reactive strategies to interrupt household-level transmission. Yet, our data also suggest that parasite populations are structured more by time than space, and therefore that household-level interventions *may not* have measurable effects on community-level risk. Thus, whether transmission foci extend into surrounding households, and to what extent mitigating them with reactive strategies contributes to a reduction in aggregate community transmission, remains to be tested by future studies.

The greatest limitation of this investigation is the lack of longitudinal sampling of cases/household members or longitudinal entomologic surveillance. We anticipate that the presence of multiple data points would enrich our understanding of parasite transmission dynamics, including the directionality and time scale of transmission, temporal fluctuations in haplotype frequency and parasite density, and the impact of parasite density upon the probability of onward transmission (and thus the clinical import of low-density infections). Another limitation is the study protocol resulted in a dataset that is (1) biased towards a young age range (given the tendency for infected children to have higher parasite density) and (2) is dominated by CC+ direct household members (CC neighbors and community members were excluded apart from control households) and thus we could not test for fine-scale decay in sharing by distance or empirically define a distance threshold for the observed genetic sharing. Additionally, our analyses of haplotype sharing between poly-genomic infections require acceptance of the assumption that parasites identical by descent can be inferred from haplotypes identical by state, and this can be undermined by convergent evolution on sequences by balancing selection and constrained haplotype frequencies across a dataset, which collectively produce identical matches by chance. These risks, however, were mitigated in our study owing to the fine temporal and spatial scales of sampling and the abundance of unique haplotypes at each locus;

in addition, the concordant findings in analyses using haplotypes derived from unlinked gene targets further suggests that matching by chance was common. Finally, as with any study requiring PCR amplification and DNA sequencing, there is the potential for systematic error (primer bias, contamination, etc) to impact the experimental findings. We reduced the risk of systematic error by enforcing strict sequencing read quality criteria, using validated haplotype inference tools, and measuring effects using unlinked polymorphic targets in orthogonal analyses of *csp* and *ama1*.

In conclusion, this is the first study to develop and apply indices of genetic similarity between infected individuals to explore dynamics of parasite transmission at both household and population levels. Collectively, our data suggest that *Pf* haplotypes are structured more clearly by time than space, but also that, at the household level, children with malaria share parasite geno-types with asymptomatically infected household members who may constitute a risk factor for childhood malaria. Subsequent longitudinal studies ought to employ repeated sampling of household members (including all age groups) as well as high-resolution entomologic surveillance. Such a study design would enable the discernment of how parasite populations change over time as well as the origin of symptomatic infections—are parasites associated with malaria disease transmitted from household members or is their origin external to the household? These definitive findings could have widespread implications for the next-generation of malaria control efforts to combat hetero-geneous transmission and ultimately to eliminate malaria disease.

## Methods

**Study design, enrollment, and sample collection.** Participants were enrolled between 18-April-2013 and 5-June-2014 as part of a case-control study[2]. Briefly, children admitted to the Webuye County Hospital with a diagnosis of malaria (confirmed by SD Bioline Pf HRP2 RDT) who resided within the six administrative sublocations immediately surrounding the hospital were eligible for enrollment as CC. Household members of CC were tested by RDT and provided a dried blood spot (DBS) within 1–7 days of enrolling the CC. At that time, CC were matched by age and village to an RDT-negative control child and all of the members of the control child's household were similarly tested and provided a DBS. A household was defined in this study as all family members and individuals residing under a single shared roof. All household members of case and control children were RDT-tested and DBS obtained at a single point in time immediately following child enrollment in the study. While case households were matched to control based on geographic proximity, neighbors and community members residing in close proximity to the enrolled household were not necessarily tested or sampled. All DBS were stored at 4 °C with desiccant, then shipped to Duke University (Durham, NC, USA) for subsequent analysis. Consent was obtained from all participants prior to enrollment. The study protocol and consent procedures were reviewed and approved by the Moi University Institutional Research and Ethics Committee (IREC/2013/13) and the Duke University Institutional Review Board (Pro00044098).

**DNA isolation and quantitative PCR.** A single 6 mm diameter punch from each DBS was deposited into a unique well of a deep 96-well plates *Genomic* DNA (gDNA) was extracted from each using a Chelex-100 protocol[46] and reconstituted in 100 μL nuclease free water. We estimated parasite densities in all RDT-positive infections using a real-time PCR assay targeting the *pfr364* motif in the parasite genome[47]. gDNA from each sample was tested in duplicate, and densities were estimated from standard curves on each reaction plate computed from amplifi-cation of a series of quantitative standards ranging from 1 to 2000 parasites/uL of whole blood.

**csp/ama1 amplification, library preparation, and sequencing.** Parasites were genotyped from each DBS obtained from RDT+ participants. Dual-indexed libraries for each target were prepared using a nested PCR strategy and then pooled for NGS on an Illumina MiSeq platform. Polymorphic segments of *P. falciparum* genes encoding circumsporozoite protein (*csp*; 288 bp) and apical membrane antigen 1 (*ama1*; 300 bp) were amplified in separate reactions from gDNA using primers that each included an identical overhang sequence (Supplementary Table 4). PCR1 reactions consisted of 3 μL of template gDNA (based on extraction ratios, this is anticipated to be approximately 0.3 μL of whole blood), 1.5 μM of each primer, 2 mM of MgCl$_2$, 300 μM each dNTP, 0.5 units of KAPA HiFi HotStart Taq (Roche), and nuclease-free water to a total reaction volume of 25 μL; cycling conditions were 95C x 3′ → (98C × 20s → 62C × 15s → 72C × 25s) × 35 → 72C ×

1′. PCR1 products were used as the template for PCR2 reactions, which used forward and reverse primers that annealed to PCR1 overhang sequences and also contained a MiSeq adaptor and a unique 8-mer index sequence (Supplementary Table 5). PCR2 reactions consisted of 1.5 µL of template, 200 nM of each primer, 1.5 mM of MgSO$_4$, 200 µM each dNTP, 0.1 units of Platinum Taq High Fidelity (Thermo-Fisher Scientific), and nuclease-free water to a total reaction volume of 25 µL; cycling conditions were 94C x 2′ → (94C × 15s → 72C × 10s → 68C × 30s) × 15 68C × 5′. PCR2 products were amplicons including requisite adaptors for MiSeq sequencing as well as unique combinations of two MiSeq indices to enable dis-aggregation of reads by sample after sequencing. Separate library pools were pre-pared for *csp* and *ama1* targets by combining an equal volume of PCR2 products from each reaction. Libraries were purified and concentrated with ethanol, electrophoresed in a 2% agarose gel, gel-purified using QIAquick gel extraction kit (Qiagen), and subsequently cleaned using Ampure XP beads (Agencourt). Finally, *csp* and *ama1* pools were combined in equimolar fashion into a single sequencing pool. The resulting single pool was divided between two MiSeq (v3 300-cycle PE) runs.

**Haplotype inference**. *csp* and *ama1* haplotypes were inferred from Illumina sequence reads. First, using Trimmomatic[48], CutAdapt[49], and BBmap[50], reads were mapped to the *P. falciparum* strain 3D7 reference sequences for *csp* or *ama1*, then primer sequences (and nucleotides directly adjacent) were trimmed from the sequences. For quality filtering of mapped reads, we used a sliding window to remove reads if the average Phred quality score over four adjacent nucleotides was <15. These quality-filtered reads were input into DADA2 (version 1.8) in order to join paired-end reads, call haplotypes, and remove chimeras[17]. Within DADA2, read quality was further enforced prior to haplotype inference using the Phred quality score for each read to model error frequency, removing reads with greater than the predicted errors. To ensure only high-quality haplotypes were included in the analysis dataset, we censored haplotypes supported by fewer than 100 reads or present in less than 1% of the data.

**Amplicon variation and nucleotide diversity**. Variation across the amplicon was assessed by Shannon Entropy (*Hn*), which is a unitless measure of variability calculated as

$$Hn = -\sum_{i=0}^{n} p_i \log_e p_i, \tag{1}$$

where *n* is the number of possible nucleotides at each position (A,C,T,G), and *p* the frequency of the variant nucleotides being compared. The sum of Shannon entropy at each nucleotide position represents the composite entropy score for the full amplicon. Nucleotide diversity (*π*) was computed as the average distance between each possible pair of sequences using[51]

$$\pi = \frac{\sum_{i}^{H} \sum_{j \leq i}^{H} d_{ij} f_i f_j}{L * N(N-1)/2}, \tag{2}$$

where *L* is the sequence length in nucleotides for *π*, *N* is the total number of reads per participant, $d_{ij}$ is the number of nucleotide differences between haplotype *i* and *j*, and $f_i/f_j$ is the number of reads belonging to haplotype *i*. $\pi_S$ and $\pi_N$ were calculated as the average *ds* and *dN* between pairs of haplotypes weighted by the haplotype's abundance:

$$\pi_{S,N} = \frac{\sum_{i}^{H} \sum_{j \leq i}^{H} d_{S_{ij},N_{ij}} f_i f_j}{L * N(N-1)/2}, \tag{3}$$

where $d_{sij}$ is *dS* between haplotype *i* and *j* sequences and $d_{Nij}$ is *dN* between haplotype *i* and *j* sequences. *π*, $\pi_S$, and $\pi_N$ were compared by Friedman test + posthoc Wilcoxon Signed-Rank test (two-tailed). Of note, *csp* amplicon sequence variability (Supplementary Fig. 4a, b) and the proportion of nucleotide diversity (Π) that occurs at nonsynonymous nucleotide positions ($\Pi_N$) are consistent with previous reports[12].

**Development of genetic similarity metrics**. To quantify the degree of genetic similarity between individuals or groups, we defined two novel metrics termed binary sharing and proportional sharing. Binary sharing (possible values: 1 = true, 0 = false) was defined as the presence in any two sampled individuals of at least one identical haplotype. Proportional sharing (continuous; possible values 0.0–1.0) was defined as the proportion of identical haplotypes shared between any two sampled individuals (A and B):

$$PHS_{AB} = \frac{A \cap B}{A \cup B}. \tag{4}$$

Complementary to these, to assess the genetic distance across samples, we utilized the unitless L1 and L2 norm:[45]

$$L_1 = \sum_{k=0}^{N} \sum_{i=0}^{n} |p_i - q_i|, \tag{5}$$

$$L_2 = \sum_{k=0}^{N} \sqrt{\sum_{i=0}^{n} (p_i - q_i)^2}, \tag{6}$$

where *N* is the sequence length in nucleotides, *n* the number of possible nucleotides at each position (A,C,T,G), and $p_i/q_i$ the frequencies of the variant nucleotides being compared. To calculate genetic similarity metrics between two groups, bootstrap iterations of each metric were performed upon a random sampling of pairs of individuals selected from each group (with replacement) and the average computed (10,000 iterations for binary/proportional sharing and 100 iterations for L1/L2 given the computational requirements). L2 was not utilized for analysis in this study because L1 and L2 are directly proportional and nearly perfectly correlated at both *csp* (Supplementary Fig. 8a; *ρ* = 1.0, Spearman Rank test) and *ama1* (Supplementary Fig. 8b; *ρ* = 0.99, Spearman Rank test). Binary/proportional share scores and the L1 norm were validated by assessing the mean values for each household (3+ members), sub-location, location, and the overall population for both *csp* and *ama1* (Supplementary Fig. 8c–h). On average, as genetic similarity metrics are compared between larger and larger groupings of study participants, the median decreases (increases for L1 norm) and there is enhanced central tendency as the range narrows to approximate the computed metric for the overall population. The calculated genetic similarity metrics for each group were compared by Kruskal Wallis test + posthoc Mann–Whitney U test (two-tailed).

**Temporospatial structure of genetic similarity**. To assess overall temporal structure of genetic similarity, we computed mean binary sharing, proportional sharing, and the L1 norm metrics between each pairing of study months (*n* = 15, yielding 120 pairwise comparisons). These metrics were computed by iterative random sampling (with replacement) of individuals between months as described above. Likewise, we assessed spatial structure by calculating each genetic similarity metrics between each pairing of administrative locations (*n* = 5, yielding 15 pair-wise comparisons) for a restricted time window (April–June 2013 or April–June 2014) by repetitive random sampling of individuals from each administrative location. A restricted time window was utilized to limit the impact of time as a confounding factor that might obscure spatial structuring of haplotypes. We compared metrics computed between pairs of months or pairs of administrative locations using a nonparametric Mann–Whitney U test, with a *p*-value <0.05 considered to be significant.

We limited assessment of spatial structure to inter-household distance owing to: (1) prior mapping of larval sites around a subset of these households, which indicate that larval sites are numerous, small, and transient[2]; (2) vector behavior in western Kenya, which occurs after 9 pm[52] and thus renders the household as epicenter of parasite transmission; and (3) the absence in the very circumscribed study site (~100 km$^2$) of other candidate features (e.g. rivers, lakes, mountain ranges).

**Genetic similarity between CC and households**. We limited the assessment of genetic similarity between case child (CC) and CHM to the subset of households with parasite sequence data from three or more infected members such that CC sharing could be assessed against at least 2 CHM for each household. We enforced this constraint to make the analysis more conservative by mitigating the risk of sampling error resulting in spurious findings regarding genetic similarity between CC and a single household member. This subset consisted of 41 households for *csp* and 45 households for *ama1*. We matched each of these households to one other household based on sampling date using a weighted scale (5 points for ±5 days, 4 points for ±10 days, 3 points for ±15 days, 2 points for ±30 days, and 1 point for ±60 days). In the event of a tie matching score, we randomly selected a single household. All households were successfully matched to at least one other household ±30 days. Furthermore, alternative matching algorithms employing case child age and household geographic location were also trialed and produced similar results.

Next, for each CC (*n* = 41 for *csp* and *n* = 45 for *ama1*) we computed the genetic similarity between CC and CHM from their *household of origin* as well as CHM from the *matched household*. Each comparison relied upon random, iterative sampling of CHM as previously described. The outputs of this process were pairwise (between CC and CHM) estimates of the likelihood of CC parasite genetic similarity with members of their own household in comparison to members of a matched household. Finally, we compared haplotype genetic similarity metrics of CC with origin CHM to those of CC with matched household members by Wilcoxon Signed-Rank test (two-tailed).

**Predictive value of CC:household genetic similarity**. We estimated the value of these genetic similarity metrics to predict temporal and geographic order in the dataset within the 38 households with *csp* and *ama1* haplotypes representing 3+ household members. *csp* and *ama1* genetic similarity metrics (binary sharing, proportional sharing, L1 norm) were calculated for the pairing of every case with household members from each case household. Values for *csp* and *ama1* were normalized and combined into binary sharing, proportional sharing, and L1 norm composite scores. On the basis of these composite scores, we rank-ordered households for each case child. Subsequently, we computed the geographic and

temporal distance between each case child and each rank-ordered household. Of note, the Fisher's Exact test comparison of observed vs. expected frequency in Table 3 is based upon the assumption that cases are distributed evenly throughout time and across geographic space.

**Genetic similarity between infections in CC.** We analyzed the genetic relatedness between infections in all passively detected CC by computing pairwise genetic similarity metrics for each unique pairing of infections ($n = 283$ for *csp* yielding 40,044 pairwise comparisons; $n = 288$ for *ama1* yielding 41,472 pairwise comparisons). We analyzed each pairwise estimate as a function of temporal distance (expressed as the interval in year between the dates of collection) or geographic distance, (expressed in kilometers), and modeled the relationships between these metrics using locally estimated scatterplot smoothing (LOESS).

**Interrogation of haplotypes for enhanced prevalence by age and in CC.** We computed the prevalence of each unique *csp* ($n = 120$) and *ama1* ($n = 180$) haplotype by month between individuals ≤5 years vs. >5 years old, defined as age $PD_H$:

$$csp \; age \; PD_{H_i} \forall_{1 < i < 120} = P_{H_{i, \leq 5y}} - P_{H_{i, > 5y}} \tag{7}$$

$$ama1 \; age \; PD_{H_j} \forall_{1 < j < 180} = P_{H_{j, \leq 5y}} - P_{H_{j, > 5y}} \tag{8}$$

where $P_{H_{i, \leq 5y}}$ is the prevalence of *csp* haplotype $i$ during a given month in individuals ≤5 years and $P_{H_{i, > 5y}}$ is the prevalence of that *csp* haplotype $i$ during a given month in >5 years (with analogous calculations for *ama1* haplotype $j$). Furthermore, we assessed haplotype prevalence in CC and in asymptomatic CHMs, and used these to calculate the PD of each haplotype as CC:CHM $PD_H$:

$$csp \; CC:CHM \; PD_{H_i} \forall_{1 < i < 120} = P_{H_{i, CC}} - P_{H_{i, CHM}} \tag{9}$$

$$ama1 \; CC:CHM \; PD_{H_j} \forall_{1 < j < 180} = P_{H_{j, CC}} - P_{H_{j, CHM}} \tag{10}$$

where $P_{H_{i, CC}}$ is the prevalence of *csp* haplotype $i$ during a given month in CC and $P_{H_{i, CHM}}$ is the prevalence of that *csp* haplotype $i$ during a given month in CHM (with analogous calculations for *ama1* haplotype $j$). We compared within each month the $PD_H$ for each haplotype using a Fisher's Exact test (two-tailed); for these comparisons, we applied a Bonferroni correction for the alpha as $p = 0.05/505$ observations ~1e-4. We used these $PD_H$ values to identify haplotypes that were significantly over-represented in specific age groups or in CC compared to CHM. This investigation served as a basis to empirically define outbreak cases of parasites bearing either *csp* or *ama1* haplotypes. We assessed spatial structure of outbreak cases vs. non-outbreak children using a purely spatial Bernoulli model in SaTScan (version 9.4.4)[53].

**Statistical analysis.** All statistical tests were two-tailed. Details of individual statistical tests are described in relevant sections throughout the methods.

**Reporting summary.** Further information on research design is available in the Nature Research Reporting Summary linked to this article.

## Data availability
Haplotype sequences are available in NCBI GenBank under accessioning numbers MK933826–MK933945 (*csp*) and MK933946–MK934125 (*ama1*). Source data is available for download from GitHub: https://github.com/codynelson08/MESAseq_compiled_code.

## Code availability
All analyses were completed in R version 3.5.1. Code for analyses is available for download from GitHub: https://github.com/codynelson08/MESAseq_compiled_code. Raw analysis results available upon request.

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

## Acknowledgements

The authors would like to acknowledge the research assistants who completed the RDT-testing and data/sample collection for this study. Furthermore, the authors would like to thank Katia Koelle (Emory University), Brandt Levitt (University of North Carolina), and Amy Wesolowski (Johns Hopkins University) for helpful discussions of analyses detailed in this manuscript, as well as Assumpta Nantume and Verona Liao (both Duke University) for assistance in the laboratory. This work was supported by NIH/NIAID R21 to W.P.O. (R21AI110478–02), and the original field study was funded by a grant from the Malaria Eradication Scientific Alliance (MESA). The funders had no role in study design, data collection and interpretation, decision to publish, or the preparation of this manuscript. The content is solely the responsibility of the authors and does not necessarily represent the official views of the National Institutes of Health.

## Author contributions

W.P.O., A.A.O. and J.M. designed and carried out field study; B.F. processed and sequenced samples; C.S.N., K.M.S. and J.S. analyzed data; S.M.T. and W.P.O. contributed expertise; and C.S.N., S.M.T. and W.P.O. wrote the paper.

## Competing interests

The authors declare no competing interests.
