## [Peer Review File · Nature Communications]

Reviewers' Comments:

Reviewer #1:

Remarks to the Author:

In this manuscript, Nelson and co-authors explore genetic signals within a large dataset of malaria-infected samples from an epidemiological study conducted in 2013/2014 in Kenya. This study is notable because it applies a still-uncommon, high-resolution form of genotyping appropriate for polyclonal infections from a high transmission setting, with high resolution spatial and temporal data, to reveal novel insights about the structure of malaria parasite genetic variation over space and time. The manuscript explore different metrics for characterizing genetic similarity between infections represented by different samples, and also reveals several important new insights that could inform malaria control policies locally and on a larger geographic scale. Specifically, the authors found that symptomatic children usually have infections that are genetically similar to asymptomatic infections from the same household, and that parasite populations are highly temporally structured in this setting, but not strongly spatially structured on a scale of 20 kilometers.

Because of the novel technical and biological components of this work, it will be of interest to a wide audience of genetic epidemiologists and inspire other studies. Though the manuscript is well written and figures are generally clear, it could be improved through attention to the following issues:

- 1) Line 196: Are there other reasonable spatial structure hypotheses to test beyond inter-household distance? For example, are there town centers where humans aggregate, or features like breeding sites that could structure the mosquito population, or other features that could be meaningful spatial factors in disease transmission at this scale?
- 2) Line 230: It's not clear to me how outbreak vs. non-outbreak haplotypes were defined formally. Further, Figure 6E is confusing. Does this result imply co-transmission of highly polyclonal infections? Some further defining details and commentary on the implications would be appreciated.
- 3) Line 265: Is there any consistent difference in haplotypic diversity/MOI between CC and household members within houses? Could this be used to infer directionality of transmission?
- 4) Line 342: Further commentary on the redundancy vs. complementarity of these metrics would be helpful. Following this exploration of their information content, is it useful to continue applying all three metrics? L1 norm gives very different signals in Figure 3. What does this mean?
- 5) How does subject age affect or not affect the genetic observations shown? Was there any evidence of haplotype stratification by parasite age, or child/adult classifications?
- 6) Figure 2: How appropriate a metric is sequencing reads for inclusion in this type of figure? For example, total read coverage probably varied between MiSeq runs; were read counts normalized by positive control samples of known parasite DNA concentration? Would such controls support the reliability of read counts for samples of widely varying parasitemia? It could be more appropriate to represent the number of samples in which haplotypes were observed rather than cumulative read count, if these issues are not fully resolved.

Line 97: Slightly confusing to say that mean log density, rather than just parasite density, was 2 orders of magnitude different.

Line 140/141: 'Collectively' appears in this sentence twice.

Reviewer #2:

Remarks to the Author:

This is a high quality manuscript that examines malaria transmission dynamics using next generation sequence of parasites collected in a cohort of individuals residing in Western Kenya. The authors show that by comparing genetic similarity between infected individuals using different methodologies that symptomatic children share infections with household members and that they can track an epidemic retrospectively even in this high transmission setting.

Major comments

1. There is a fundamental issue related to sensitivity of the approach which is think is confounded by both the relatively limited age range (even the household members table 1 there appear to be no adults) and the methodology used to examine parasite populations. Essentially the chance of detecting a parasite and therefore a genetic signature to work with is totally dependent on parasite density. These will be higher in children yet infections in the population as have been reported by these authors and others predominate at much lower levels – typically below the threshold for RDT. Moreover, the ability to detect infections depends not only density but how much sample is amplified, so in the methods a 6mm spot (line 406 how much blood?) goes into 100ul but only 3ul of this is used – this is a very small blood volume to work from. The authors detail failure to amplify in figure 1 which further reduces the number of samples to work with by about 30% - which statistically speaking means any observations vary +/-30%. So, whilst I think the observations of the authors about household similarity between cases and asymptomatic infections stand (this has been reported empirically before by themselves and others) these are limited and this needs to be discussed more. Has any formal sensitivity analysis been done to examine the effect of varying sensitivity of molecular detection or extending the age range ? It is very likely that there are other parasites in circulation that are not being detected in the high density samples.
2. The authors themselves acknowledge the limitation of the sampling approach with regard to adjacent households (line 351) and as above I wonder if some sort of sensitivity analysis could be conducted to assess how robust observations are given this sampling approach. The concentration of infections in a homestead (which I presume this is the case in Kenya as opposed to the more nuclear villages in West Africa – it would be helpful to clarify this). It would be helpful to complete the explanation that the comparisons were influenced by time rather than space as its not immediately intuitive.
3. Related to the above it would also be important to clarify why similarity data (methods line 512-onward) were limited to households with 3 or more samples. Was any analysis done on households with lower numbers of samples?
4. I checked the document and only found the word anopheles mentioned once (no mosquito) in relation to flight distance (line 283 which was a commented as perplexing as infections were 20km apart but surely people move much further than mosquitoes?) . Mosquito behaviour has much more impact than simple distance abundance/density, seasonality, frequency of biting, age distribution of bites etc could all influence the factors described and deserve some discussion.

Minor comments

1. Intro line 54 – 58 these gaps are acknowledged but we do have information on basic parasite and mosquito biology. I think the issue is more the dynamics of natural infections and how this influences transmissibility.
2. Intro 78 – spatial aggregation is presumably associated with environmental factors not just asymptomatic carriers alone – the risk is being in a malarious area/household
3. Results 105 – I may have missed this but how many had AMA and CSP haplotypes
4. Results 141 – the alternative explanation in line with point 1 above is that the sensitivity of the

method and the sampling do not allow detection of any spatial structure

5. Results 187 – consider rephrasing sentence ` An index

6. Results 229 – check date formats throughout – there 3 different variants

7. Discussion 269 – I think this is an interesting concept but screen & treat approaches have largely failed. It will depend on the sensitivity of diagnostic and transmission level. In studies to date several those parasitaemic at the time where the first to be reinfected confirming risk is spatial.

26 July 2019

Point-by-point responses to reviewer comments

Nelson et al., “High-resolution micro-epidemiology of parasite spatial and temporal dynamics in a high malaria transmission setting in Kenya”

Original reviewer comments are numbered and in italics, and our responses are indented and in normal font. Additionally, changes to the text are indicated throughout this document and the manuscript in blue text.

Reviewer #1:

1) *Line 196: Are there other reasonable spatial structure hypotheses to test beyond inter-household distance? For example, are there town centers where humans aggregate, or features like breeding sites that could structure the mosquito population, or other features that could be meaningful spatial factors in disease transmission at this scale?*

RESPONSE: The lack of spatial structuring (except within individual households) identified in our investigations was quite surprising to us. As suggested by the reviewer, we contemplated assessing alternative spatial hypotheses including population density, topography, and geographic features that might explain our results. However, we decided to defer any formal assessment apart from household location since the vast majority of infectious anopheline vector biting occurs after 9pm, and thus the household constitutes the epicenter of transmission. Larval sites were catalogued and mapped in this study, and were identified to be small, local, and seasonal. Overall, however, there was limited entomological surveillance. Notably, there is not any single, large geographic feature that might serve as a source of mosquitoes and create vector structure in space (river, lake, etc.), and thus we could not assess any relationship of household proximity to breeding sites with transmission intensity and/or haplotype identity. These analytic choices are now described in the methods section (page 25, line 551-556).

→ “We limited assessment of spatial structure to inter-household distance owing to: 1) prior mapping of larval sites around a subset of these households, which indicate that larval sites are numerous, small, and transient [Obala et al., *Plos One*, 2015]; 2) vector behavior in western Kenya, which occurs after 9pm [Wamae et al., *Acta Trop*, 2015] and thus renders the household as epicenter of parasite transmission; and 3) the absence in the very circumscribed study site (~100 km²) of other candidate features (e.g. rivers, lakes, mountain ranges).”

2) *Line 230: It’s not clear to me how outbreak vs. non-outbreak haplotypes were defined formally. Further, Figure 6E is confusing. Does this result imply co-transmission of highly polyclonal infections? Some further defining details and commentary on the implications would be appreciated.*

RESPONSE: We have clarified this analysis in the results section, detailing the step-by-step procedures that led to the identification of outbreak cases and including a formal case definition (page 11, line 234-242).

→ **Results:** “Comparing the monthly PD_H of each haplotype, we determined that 4 haplotypes (*csp* H8, H48, and H54 as well as *ama1* H13) were significantly more common in CC than CHM during June 2013 (each $p < 0.0001$ by Fisher Exact test) (**Figure 6a,b**). We

examined CC from 5/6/2013–7/29/2013, noting all those infected with parasites bearing *csp* H8, H48, and H54 also had evidence of *csp* H1 (Figure 6c). Likewise, from 5/6/2013–7/29/2013 all CC in which *ama1* H13 was detected also had *ama1* H5 and H8 (Figure 6d). In total we identified 26 CC with *csp* H1 + H8 + H48 + H54 and 27 with *ama1* H5 + H8 + H13 (Fig. 6c,d). Intriguingly, we observed substantial overlap of this set of haplotypes in case children: 23 CC had evidence of all haplotypes combined (*csp* H1 + H8 + H48 + H54 and *ama1* H5 + H8 + H13) (Fig. 6e) [...] We defined an outbreak ‘case’ as the presence of 5 or more of the 7 outbreak haplotypes (*csp* H1/H8/H48/H54 and *ama1* H5/H8/H13), comprising more than 98% of the reads detected in an individual. Employing this definition, we identified a total of 29 outbreak cases and 48 non-outbreak cases among the 77 total CC between 5/6/2013 and 7/29/2013.”

Furthermore, we have clarified our empiric definition of outbreak vs. non-outbreak haplotypes in the methods section (page 27, line 618-622).

→ “We used these PD_H values to identify haplotypes that were significantly over-represented in CC compared to CHM, which served as a basis to empirically define outbreak cases of parasites bearing either *csp* or *ama1* haplotypes. We assessed spatial structure of outbreak cases vs. non-outbreak children using a purely spatial Bernoulli model in SaTScan (version 9.4.4).”

The means by which this co-occurrence of a unique combination of haplotypes within a brief temporal window occurs is unclear to us, as indicated within the discussion section (page 14, line 300-302): “The mechanism of this long-range transmission (cases up to 20km apart) of genetically-identical infections is perplexing, since the flight range of unfed *Anopheles gambiae* has been measured at a maximum of 3km.” We believe that the identification of ‘outbreak’ cases amidst endemic malaria transmission may be contingent upon unconventional/undescribed vector movement and biting behavior. We have expanded upon our uncertainty regarding the origin of these results in the discussion section (page 15, line 319-325).

→ “The reason for the co-occurrence of this unique combination of haplotypes among symptomatic children is unclear, and two major questions remain unanswered by this investigation: 1) How did the outbreak spread nearly simultaneously across a relatively large geographic area and 2) why was this combination only detected among CC? The geographic co-occurrence of outbreak cases may be contingent upon unconventional and/or undescribed vector movement and biting behavior or cryptic human movements within the study area.”

3) Line 265: Is there any consistent difference in haplotypic diversity/MOI between CC and household members within houses? Could this be used to infer directionality of transmission?

RESPONSE: Of note, there is no difference in median MOI between CC and CHM at the population level for both *csp* and *ama1* haplotypes (Mann-Whitney U test, $p=ns$). At the reviewer’s suggestion, we investigated whether there is any pairwise difference in MOI within a single household and identified no systematic difference between MOI of CC vs CHM detected at either *csp* or *ama1* loci (Wilcoxon Sign-Rank test, $p>0.05$). These findings have been added to the results section of the manuscript (page 7, line 126-128).

→ “However, there was no consistent difference between the MOI detected in CC vs. CHM within a single household ($p=ns$, Wilcoxon Sign-Rank test).”

We hypothesize that it is possible to infer directionality of transmission from parasite genomic data, but that such an analysis would require longitudinal sampling of individuals.

As previously mentioned in our discussion (page 18, line 390-391), the lack of longitudinal sampling is a limitation of our study that only allows us to assess genetic similarity between two persons A and B at a single point in time. Yet longitudinal sampling coupled with analysis of haplotype identity/frequency changes over time at the household level should give a clue regarding the origin of infection and directionality of transmission – whether the infection spread from A to B, from B to A, OR whether A and B acquired the infection from the same external source. This issue has been expounded upon further in the discussion (page 18, line 391-395).

→ “We anticipate that a longitudinal dataset would enrich our understanding of parasite transmission dynamics, including the directionality and time scale of transmission, temporal fluctuations in haplotype frequency and parasite density, as well as the impact of parasite density upon the probability of onward transmission [...]”

4) *Line 342: Further commentary on the redundancy vs. complementarity of these metrics would be helpful. Following this exploration of their information content, is it useful to continue applying all three metrics? L1 norm gives very different signals in Figure 3. What does this mean?*

RESPONSE: The reviewer notes that we have introduced and applied three metrics of genetic similarity (which most frequently appear to provide complementary results) without giving any guidance regarding future usage of these metrics. Let us first consider binary vs. proportional sharing as these appear the most homologous to one another (Supplementary Fig. 8). Because it measures any shared haplotypes, binary sharing most likely has higher dynamic range and enhanced sensitivity of detection though this metric has greater ‘noise’ and is highly influenced by the occurrence of common haplotypes. For example, the average *csp* binary share score for the entire population is 25, meaning that 1 in 4 random samplings of people share at least one haplotype. By comparison, the average proportional share score is 5 meaning that any randomly sampled individuals share only 5% of haplotypes. Therefore, whether one should employ binary or proportional sharing for an analysis depends upon the purpose of the investigation and/or the preference of the researcher (conservative vs. exploratory analysis), as binary sharing is almost certainly more sensitive at detecting any genetic identity and proportional sharing more specific for discerning a population-level sharing trend. We can hypothesize that proportional sharing might have greater utility when there are more haplotypes, and binary sharing when the number of haplotypes is relatively small.

We anticipate the L1 norm is somewhat different, being weighted not only by haplotype abundance but also by the degree of sequence divergence. The reviewer has picked up on a subtle finding regarding the L1 norm being heterogeneous between villages. We therefore anticipate that there may be spatial structuring of the genetic similarity identified by the L1 norm. We are not certain of the implications, though it seems evident that the L1 norm is a related yet distinct measure of genetic similarity.

At this point in time it is difficult to ascertain which genetic similarity metric ought to be used for specific purposes. We anticipate that the use of these metrics will depend upon local epidemiology, namely parasite genetic diversity, transmission intensity, and prevalence of parasitemias. Thus, subsequent simulation studies and/or application to diverse datasets will further delineate the similarities and differences between these metrics, revealing the utility of their use. However, we believe these metrics are a valuable addition to the toolkit of understanding polyclonal infections, which will enable investigators to define connectivity between parasite infections. We have added several lines of text to the discussion section describing these findings and suggestions for future use (page 16-17, line 358-356).

→ “We suggest that binary and proportional sharing metrics produce highly similar results in our analyses, whereas the L1 norm results are somewhat distinct and heterogeneous (**Fig. 3, Supplementary Figs. 7,8**). We anticipate that appropriate use of these metrics will depend on local epidemiology, namely parasite genetic diversity, transmission intensity, and prevalence of parasitemias. Thus, we propose these metrics ought to be applied to diverse datasets to define the context of their utility. Nevertheless, we hypothesize these high-resolution genetic metrics will enable investigators to identify connectivity between polygenomic infections on more granular temporal and geographic scales.”

5) *How does subject age affect or not affect the genetic observations shown? Was there any evidence of haplotype stratification by parasite age, or child/adult classifications?*

RESPONSE: Given that age (<10 years old) was an enrollment criterion for case children in this study, it is difficult to parse out any effect of age vs. clinical disease symptoms on strain haplotype/strain bias. Nevertheless, to investigate this question further, we looked for age-related haplotype bias by assessing haplotype prevalence difference for all 120 *csp* and 180 *ama1* haplotypes between young children (≤5 years old) and older children/adults (>5 years old) (analogous to the analysis done in fig 6a/b for prevalence difference between CC and CHM). There are haplotypes from a roughly equivalent number of study participants for <5 and >5 groupings (*csp*: 296 for ≤5y, 357 for >5y; *ama1*: 300 for ≤5y and 365 for >5y). Overall, there is no consistent biases towards specific strains/haplotypes within either the ≤5y or >5y groupings. However, we observed that four haplotypes (2 *csp* and 2 *ama1*) are statistically more common in the ≤5y population during 10/2013. These data are presented in new supplementary figure 6 and discussed in the results (page 7, line 139-144).

→ “We tested if haplotype presence was impacted by age, because parasite density (and thereby haplotype detection sensitivity) often depends upon host age in areas of endemic transmission [Baird et al., *Parasitology Today*, 1995]. To do so, we computed the prevalence difference of each haplotype (PD_{H_i}) between young children (≤5y) and older children/adults (>5y). However, we observed no consistent difference in haplotype prevalence between the ≤5y and >5y populations (**Supplementary Fig. 6**).”

Furthermore, details regarding this analysis was added to the methods section (page 27, line 603-609).

→ “We computed the prevalence of each unique *csp* (n=120) and *ama1* (n=180) haplotype by month between individuals ≤5 years vs. >5 years old, defined as age PD_{H_i} :

$$csp \text{ age } PD_{H_i} \forall 1 < i < 120 = P_{H_i, \leq 5y} - P_{H_i, > 5y}$$

$$ama1 \text{ age } PD_{H_j} \forall 1 < j < 180 = P_{H_j, \leq 5y} - P_{H_j, > 5y}$$

where $P_{H_i, < 5y}$ is the prevalence of *csp* haplotype *i* during a given month in individuals ≤5y and $P_{H_i, > 5y}$ is the prevalence of that *csp* haplotype *i* during a given month in >5y (with analogous calculations for *ama1* haplotype *j*).”

6) *Figure 2: How appropriate a metric is sequencing reads for inclusion in this type of figure? For example, total read coverage probably varied between MiSeq runs; were read counts normalized by positive control samples of known parasite DNA concentration? Would such controls support the reliability of read counts for samples of widely varying parasitemia? It could be more appropriate to represent the number of samples in which haplotypes were observed rather than cumulative read count, if these issues are not fully resolved.*

RESPONSE: The reviewer presents a fair critique of the data presented in Figure 2 and Supplementary figure 5. The read counts were not normalized by positive control samples of known parasite DNA concentration, and therefore we have removed this metric from the both figure 2 and supplementary figure 5.

7) *Line 97: Slightly confusing to say that mean log density, rather than just parasite density, was 2 orders of magnitude different.*

RESPONSE: The wording of this sentence has been changed as suggested by the reviewer to read: “the median parasite density was nearly 2 orders of magnitude higher for samples successfully assigned csp [...]” (page 6, line 101-103).

8) *Line 140/141: ‘Collectively’ appears in this sentence twice.*

RESPONSE: The second iteration of this word was deleted such that the sentence reads: “Collectively, these findings suggest a lack of spatial structuring of haplotypes” (page 8, line 154-155).

Reviewer #2:

9. *There is a fundamental issue related to sensitivity of the approach which is think is confounded by both the relatively limited age range (even the household members table 1 there appear to be no adults) and the methodology used to examine parasite populations. Essentially the chance of detecting a parasite and therefore a genetic signature to work with is totally dependent on parasite density. These will be higher in children yet infections in the population as have been reported by these authors and others predominate at much lower levels – typically below the threshold for RDT. Moreover, the ability to detect infections depends not only density but how much sample is amplified, so in the methods a 6mm spot (line 406 how much blood?) goes into 100ul but only 3ul of this is used – this is a very small blood volume to work from. The authors detail failure to amplify in figure 1 which further reduces the number of samples to work with by about 30% - which statistically speaking means any observations vary +/-30%. So, whilst I think the observations of the authors about household similarity between cases and asymptomatic infections stand (this has been reported empirically before by themselves and others) these are limited and this needs to be discussed more. Has any formal sensitivity analysis been done to examine the effect of varying sensitivity of molecular detection or extending the age range? It is very likely that there are other parasites in circulation that are not being detected in the high density samples.*

RESPONSE: To clarify, adults were included in the study, though they constituted a minority of the dataset. The data in table 1 as well as supplementary table 1 has been changed from median + IQR to median + range to emphasize that adults were included. That our dataset is biased towards a younger age range has been added to the discussion section as a limitation of this work (page 18, line 395-397).

→ “Another limitation is the study protocol resulted in a dataset that is 1) biased towards a young age range (given the tendency for infected children to have higher parasite density [...])”

To answer the reviewer’s question, a 6mm DBS is anticipated to be 10uL of blood so loading 3/100uL of extracted DNA is equivalent to approximately 0.3uL of whole blood. Despite the small volume of blood assayed, we have parasite density data indicating that

haplotype assignment was quite sensitive. Indeed, the widely cited sensitivity cutoff for conventional RDT tests of 100 parasites/uL represents the 38th percentile for samples assigned csp haplotypes.

The clinical significance of ultra-low density infections (ULDI, here defined as <10 parasites/uL) remains unknown. Is *in vitro* identification of ULDI a real or spurious finding? Does ULDI represent infection resolution? And what impact to ULDI have on forward transmission of parasites? These and many other questions remain to be answered. The reviewer is correct that our investigation is limited by the assumption that the properties of parasites found in ULDI do not differ from the overall pool of parasites in individuals with higher density infections, and testing this assumption has been largely unfeasible owing to the limited technical ability to obtain sequence data from such low-density infections. We suggest that future investigations utilizing a longitudinal cohort might be able to address this critical question. We have added this point to the discussion section (page 18, line 391-395).

→ “We anticipate that a longitudinal dataset would enrich our understanding of parasite transmission dynamics, including [...] the impact of parasite density upon transmission likelihood or intensity (and thus the clinical import of low-density infections).”

Finally, as suggested by the reviewer (and as described in comment #5 above), we investigated whether there is any difference in haplotypes detected in young children (≤ 5 years old, higher mean parasite density) vs. older children/adults (> 5 years old, lower mean parasite density) – see supplementary figure 6. We failed to observe any consistent difference in haplotype occurrence between the ≤ 5 and > 5 populations, which suggests to us that we are not systematically failing to detect specific populations of parasites in high-density vs. low-density infections (and vice versa). This analysis and our interpretation has been added to the results section of the manuscript (page 7, line 139-144).

→ “We tested if haplotype presence was impacted by age, because parasite density (and thereby haplotype detection sensitivity) often depends upon host age in areas of endemic transmission [Baird et al., *Parasitology Today*, 1995]. To do so, we computed the prevalence difference of each haplotype (PD_H) between young children ($\leq 5y$) and older children/adults ($> 5y$). However, we observed no consistent difference in haplotype prevalence between the $\leq 5y$ and $> 5y$ populations (**Supplementary Fig. 6**).”

10. The authors themselves acknowledge the limitation of the sampling approach with regard to adjacent households (line 351) and as above I wonder if some sort of sensitivity analysis could be conducted to assess how robust observations are given this sampling approach. The concentration of infections in a homestead (which I presume this is the case in Kenya as opposed to the more nuclear villages in West Africa – it would be helpful to clarify this). It would be helpful to complete the explanation that the comparisons were influenced by time rather than space as its not immediately intuitive.

RESPONSE: Like the reviewer, we certainly recognize that the conclusions to be drawn in this study regarding the temporality and spatiality of transmission are limited by the study design. In this study, a household was defined as all family members and individuals residing under a shared roof, and all household members were RDT-tested and DBS were collected at a single time point immediately following case child or control child enrollment in the study. While case households were matched to control based on geographic proximity, we note that neighbors and community members adjacent to enrolled household members

were not necessarily tested. These details have been added to the methods section (page 20, line 433-438).

→ “A household was defined in this study as all family members and individuals residing under a single shared roof. All household members of case and control children were RDT-tested and DBS obtained at a single point in time immediately following child enrollment in the study. While case households were matched to control based on geographic proximity, neighbors and community members residing in close proximity to the enrolled household were not necessarily tested or sampled.”

Thus, the sampling in this study was focused upon the people who share a sleeping space with children with clinical malaria disease (or control children) at a single moment in time. Though a circumferential sampling of neighbors who live proximally to the case child household would have been preferable in order to best comment upon the spatiality of transmission, this was not done and so we must interpret our findings with this caveat. This limitation is further expanded upon within the discussion section (page 18, line 395-400).

→ “Another limitation is the study protocol resulted in a dataset that [...] is dominated by CC + direct household members (CC neighbors and community members were excluded apart from control households) and thus we could not test for fine-scale decay in sharing by distance or empirically define a distance threshold for the observed genetic sharing.”

As a form of sensitivity analysis, we made several efforts to restrict the temporal window and therefore make our analysis regarding spatial structuring of haplotypes as robust as possible. First, we narrowed the investigation to a single 3-month window (Figure 3h-m, Supplementary Figure 6g-l). While binary and proportional sharing were fairly homogenous between villages, the L1 norm appears to be somewhat inconsistent (as described above). In addition to assessing 3-month slices to time, we further limited the scope to the outbreak cases identified in Figure 6 and identified no geographic clustering of these genetically-identical cases (Figure 6j). Our attempt to mitigate the impact of time upon our spatial analysis has been expanded upon in the methods section (page 24, line 547-548).

→ “A restricted time window was utilized to limit the impact of time as a confounding factor that might obscure spatial structuring of haplotypes.”

11. Related to the above it would also be important to clarify why similarity data (methods line 512- onward) were limited to households with 3 or more samples. Was any analysis done on households with lower numbers of samples?

RESPONSE: We reported an analysis of genetic similarity between case children and households that was limited to those households with 3 or more total members (i.e. index case + 2 additional members). We chose this parameter to minimize bias potentially introduced by sampling error and binary detection (+/- haplotype sharing) leading to artifactual findings and biasing the results. The more household members that are present (higher the household n), the more likely that the calculated genetic similarity indices represent the true value. Consider the following scenario in which the CC and CHM1 have the same set of haplotypes:

2-member household

CC – H5
CHM1 – H1, H8

Binary sharing = 0
Proportional sharing = 0

5-member household

CC – H5
CHM1 – H1, H8
CHM2 – H5, H8
CHM3 – H8
CHM4 – H5

Binary sharing = 50
Proportional sharing = 37.5

The threshold of 3 total household members was a compromise because it afforded us a reasonable number of households (*csp* n=41, *ama1* n=45), while limiting the chance of spurious findings for a household size of 2. More details have been added to the methods section to reduce confusion regarding this analytic choice (page 25, line 562-564).

→ “We enforced this constraint to make the analysis more conservative by mitigating the risk of sampling error resulting in spurious findings regarding genetic similarity between CC and a single household member”

At the reviewer’s suggestion, we completed an identical analysis using all households with 2+ total members (case child + at least 1 additional household member; *csp* n=84, *ama1* n=90). Our findings of enhanced sharing between CC and origin household CHM held at the *csp* locus (binary sharing p=0.04, proportional sharing p=0.01, L1 norm p=0.02) though not at the *ama1* locus (p=ns for all). We believe that this observation reinforces our findings because the balance of evidence remains in favor of enhanced sharing between CC and origin household CHM. We anticipate the failure to observe a statistically significant result at the *ama1* locus is simply due to stochastic sampling error based upon how many individuals were sequenced from within an individual household. These findings and interpretation have also been added to the results section of the manuscript (page 9-10, line 197-200).

→ “Lastly, if we lift the restriction that a household be comprised of 3+ individuals to be included in this analysis, the findings hold at the *csp* though not the *ama1* locus, possibly owing to a greater diversity of *ama1* haplotypes overall (180) compared to *csp* (120), and the resulting lower probability of observing exact matches.”

12. *I checked the document and only found the word anopheles mentioned once (no mosquito) in relation to flight distance (line 283 which was commented as perplexing as infections were 20km apart but surely people move much further than mosquitoes?). Mosquito behaviour has much more impact than simple distance abundance/density, seasonality, frequency of biting, age distribution of bites etc could all influence the factors described and deserve some discussion.*

RESPONSE: The reviewer has recognized that complex mosquito biting behavior is likely at play in temporal structuring of haplotype genetic similarity (Figure 3, Figure 5) as well as the detection of outbreak haplotypes among case children (Figure 6). Given that anopheline-mosquitoes are night-biters, we propose that human movement is only relevant in the case of overnight stay outside the home; thus, frequent, relatively short (<20km) overnight trips would be required to drive the mixing seen here. Interestingly, it was previously noted in this cohort that there was clustering of RDT+ individuals within the households of CC compared to control (Obala et al, 2015). The relatively-limited entomology in this investigation also indicated greater clustering of larval sites and fed mosquitoes in case households compared to control, though this was a less robust effect and would not explain the long-range dispersal of genetically-identical parasites. As suggested by the reviewer, we have added

several lines to the introduction (page 4, line 79-83) as well as the discussion (page 15, line 323-325) to highlight the likely role of vector behavior in these findings and gaps in knowledge regarding the entomology and mosquito biting behavior responsible for our observed results.

→ **Introduction:** “Previously, we observed clustering of RDT-positive individuals, noting that infections were 2.5 times more common among the household members of cases compared to controls [Obala et al., *Plos One*, 2015]. Though entomology in this study was quite limited, we also identified clustering of larval sites and bloodfed anopheline mosquitoes in case households; however, these relatively weak associations suggest that vector proximity is not a primary driver of disease risk.”

→ **Discussion:** “The geographic co-occurrence of outbreak cases may be contingent upon unconventional and/or undescribed vector movement and biting behavior [...] .”

13. *Intro line 54 – 58 these gaps are acknowledged but we do have information on basic parasite and mosquito biology. I think the issue is more the dynamics of natural infections and how this influences transmissibility.*

RESPONSE: The language in the introduction has been modified according to the reviewer’s suggestion (page 3, line 56-59).

→ “A greater understanding of the dynamics of natural infections and their impact on parasite transmissibility could enable rational implementation of control measures to reduce the malaria disease burden in high-transmission settings.”

14. *Intro 78 – spatial aggregation is presumably associated with environmental factors not just asymptomatic carriers alone – the risk is being in a malarious area/household*

RESPONSE: As recommended, this statement has been modified slightly to indicate that spatial aggregation of cases likely occurs regardless of whether household members are symptomatic or asymptomatic (page 4, line 79-80).

→ “Previously, we observed clustering of RDT-positive individuals, noting that infections were 2.5 times more common among the household members of cases compared to controls [Obala et al., *Plos One*, 2015].”

15. *Results 105 – I may have missed this but how many had AMA and CSP haplotypes*

RESPONSE: This information has been added to the results section (page 6, line 100-101).

→ “In total, 617 samples (64% of infections initially submitted for sequencing) were assigned both *csp* and *ama1* haplotypes.”

16. *Results 141 – the alternative explanation in line with point 1 above is that the sensitivity of the method and the sampling do not allow detection of any spatial structure*

RESPONSE: This point has been added to the discussion section of the manuscript within a paragraph describing the limitations of our study/findings (page 18, line 395-400).

→ “Another limitation is the study protocol resulted in a dataset that [...] is dominated by CC + direct household members (CC neighbors and community members were excluded apart from control households) and thus we could not test for fine-scale decay in sharing by distance or empirically define a distance threshold for the observed genetic sharing.”

17. Results 187 – consider rephrasing sentence ‘ An index

RESPONSE: As recommended by the reviewer, we have changed the text to improve the coherence of this sentence (page 10, line 205-208).

→ “We surmised that if binary sharing, proportional sharing, and the L1 norm are highly-predictive indices, the calculated CC:CHM value should be greatest for the comparison of CC with their own household members thus accurately identifying the CC household of origin.”

18. Results 229 – check date formats throughout – there 3 different variants

RESPONSE: We have thoroughly reviewed the text as well as the figures and changed all variant dates to match the format that appears on page 11, line 236 (MM/DD/YYYY).

19. Discussion 269 – I think this is an interesting concept but screen & treat approaches have largely failed. It will depend on the sensitivity of diagnostic and transmission level. In studies to date several those parasitaemic at the time where the first to be reinfected confirming risk is spatial.

RESPONSE: We agree with the reviewer’s comment that infections/re-infections have been observed to cluster spatially. In this same cohort of individuals, the median number of infections was more than double in case households compared to control (Obala et al., Plos One, 2015) – i.e. being in a malarious household increased the risk for clinical malaria disease dramatically. Our data goes one step further and identifies that infected individuals within a household have enhanced parasite genetic similarity, which lends weight to the hypothesis that infected members are a risk factor for other household members to develop disease. However, there were also unique haplotypes within households indicating that intra-household sharing was not the only source of new infections. Overall this suggests that reactive case detection (RCD) approaches could reduce household-level transmission, but will not eliminate the risk of new infections. Our results are consistent with the results of RCD studies in high transmission settings – 1) intra-household genetic identity is an important risk factor but not the exclusive source of new infections which limits the impact of household RCD and 2) lack of spatial structure and high degree of parasite sharing beyond the household level indicates that household-level interventions may not have measurable effects on community-level infection risk. However, as pointed out by the reviewer, diagnostic test sensitivity and transmission intensity might change the effectiveness of screen and treat approaches, and ought to be studied in subsequent investigations. We have added these points to a paragraph discussing the implications of our findings (page 17, line 382-389).

→ “This investigation provides direct evidence that clinically-silent parasite transmission chains within households are an important risk factor (but not the exclusive source) of new infections, which supports the rationale for employing reactive strategies to interrupt household-level transmission. Yet, our data also suggests that parasite populations are structured more by time than space, and therefore that household-level interventions may not have measurable effects on community-level risk. Thus, whether transmission foci extend into surrounding households, and to what extent mitigating them with reactive strategies contributes to a reduction in aggregate community transmission, remains to be tested by future studies.”

Reviewers' Comments:

Reviewer #1:

None

Reviewer #2:

Remarks to the Author:

This is a comprehensive response to the reviewers comments and I have a few minor comments

I do think the lack of entomology is a limitation and needs to be highlighted a little more - the reference they cite (Wamae) actually focuses on early biting as a risk for transmission and success of interventions.

At the very least the recommendation for longitudinal studies should include complimentary entomology studies.

Similarly this longitudinal study should include individuals of all ages.

I could not find if the clarification of the actual blood volume equivalent is now include in the text. This is important for reproducibility

I do not disagree with premise but the Baird et al 1995 ref is a bit old – something more current that reflects the relationship with current interventions would be more appropriate - there are several out there many which include actual multiplicity of infection

Second-round reviewer comments

Reviewer #2:

This is a comprehensive response to the reviewers' comments and I have a few minor comments

I do think the lack of entomology is a limitation and needs to be highlighted a little more - the reference they cite (Wamae) actually focuses on early biting as a risk for transmission and success of interventions.

We have added the limited entomology as a limitation in the discussion as suggested (page 18, line 394-396). Furthermore, in the results we have included an alternative and more appropriate references to the Wamae paper (Bayoh, *Parasite Vectors*, 2014).

At the very least the recommendation for longitudinal studies should include complimentary entomology studies.

As suggested by the reviewer, we had added in the recommendation that longitudinal studies ought to incorporate entomology and entomologic outcomes (page 19, line 421-424).

Similarly this longitudinal study should include individuals of all ages.

We have included the recommendation that future studies ought to include individuals of all ages (page 19, line 421-424).

I could not find if the clarification of the actual blood volume equivalent is now include in the text. This is important for reproducibility.

The reviewer is correct that this information was not added to the text of the manuscript itself, but only in the point-by-point response. We have added this information to the text as suggested (page 21, line 463-464).

I do not disagree with premise but the Baird et al 1995 ref is a bit old – something more current that reflects the relationship with current interventions would be more appropriate - there are several out there many which include actual multiplicity of infection

We have added a more current reference regarding this topic – Rodriguez-Barraquer et al., *eLIFE*, 2018.